# RealCompo: Balancing Realism and Compositionality Improves Text-to-Image Diffusion Models

**Xinchen Zhang**[1*]   **Ling Yang**[2*]   **Yaqi Cai**[3]   **Zhaochen Yu**[2]   **Kai-Ni Wang**[4]
**Jiake Xie**[5]   **Ye Tian**[2]   **Minkai Xu**[6]   **Yong Tang**[5]   **Yujiu Yang**[1†]   **Bin Cui**[2†]

[1]Tsinghua University   [2]Peking University   [3]University of Science and Technology of China
[4]Southeast University   [5]LibAI Lab   [6]Stanford University
https://github.com/YangLing0818/RealCompo

## Abstract

Diffusion models have achieved remarkable advancements in text-to-image generation. However, existing models still have many difficulties when faced with multiple-object compositional generation. In this paper, we propose ***RealCompo***, a new *training-free* and *transferred-friendly* text-to-image generation framework, which aims to leverage the respective advantages of text-to-image models and spatial-aware image diffusion models (e.g., layout, keypoints and segmentation maps) to enhance both realism and compositionality of the generated images. An intuitive and novel *balancer* is proposed to dynamically balance the strengths of the two models in denoising process, allowing plug-and-play use of any model without extra training. Extensive experiments show that our RealCompo consistently outperforms state-of-the-art text-to-image models and spatial-aware image diffusion models in multiple-object compositional generation while keeping satisfactory realism and compositionality of the generated images. Notably, our RealCompo can be seamlessly extended with a wide range of spatial-aware image diffusion models and stylized diffusion models.

## 1   Introduction

The field of diffusion models has witnessed exciting developments and significant advancements recently[65, 46, 19, 45, 40, 73]. Among various generative tasks, text-to-image (T2I) generation [33, 20, 64] has gained considerable interest within the community. T2I diffusion models such as Stable Diffusion [41], Imagen [42] and DALL-E 2/3 [39, 4] have exhibited powerful capabilities in generating images with high aesthetic quality and realism [4, 36]. However, they often struggle to align accurately with the compositional prompt when it involves multiple objects or complex relationships [28, 3, 34], which requires the model to have strong spatial-aware ability.

One potential solution to optimize the compositionality of generated images is providing a spatial-aware condition to control diffusion models [12, 66, 58], such as layout/boxes [35, 14], keypoint/pose [72] and segmentation map [22]. These spatial-aware conditions are fundamentally similar in functioning, thus we mainly focus our analysis on layout-to-image (L2I) models for simplicity. With the control of layout, L2I models [27, 8, 59] improve compositionality by generating objects at specified locations. For instance, GLIGEN [27] designs trainable gated self-attention layers to incorporate layout input and controls the strength of its incorporation by changing parameter $\beta$. Although L2I models improve the weaknesses of compositional text-to-image generation, their generated images exhibit a significant decline in realism compared to T2I models [27, 78].

---

*Contributed equally.
†Corresponding authors.

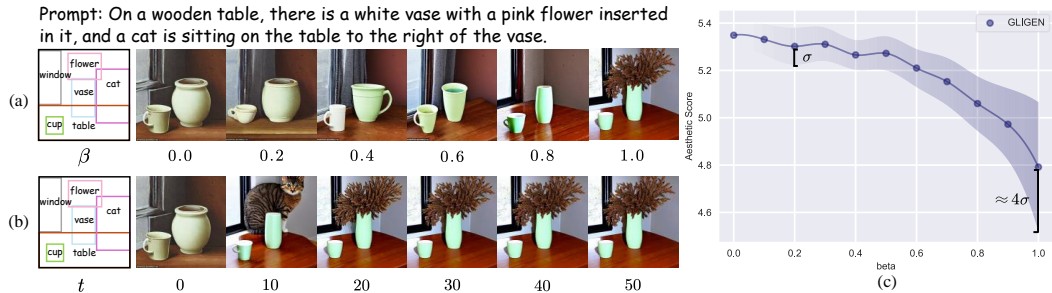

Figure 1: **Motivations of RealCompo**. **(a)** and **(c)** The realism and aesthetic quality of generated images become poor as more layout is incorporated. **(b)** Even if layout is incorporated only in the early denoising stages, the control of text alone still fails to alleviate the poor realism issue. More results are shown in Appendix B.

We conducted experiments to analyze why a significant decrease in image realism exists. We analyze the layout injection mechanism in GLIGEN [27] by controlling the density of layout through parameter $\beta$. As shown in Fig. 1 (a) and (c), our experiments indicate that the density of layout directly influences the realism of generated images. As the control of layout gradually increases, the generated images become less aesthetic and more unstable. This demonstrates that layout and text, as different control conditions, guide the model towards different generation directions, with the former emphasizing compositionality and the latter emphasizing realism. To alleviate this issue, some models [28, 27] leverage the early-stage localization capability of diffusion models [71, 49] and incorporate layouts only during the initial denoising phase. In the later denoising stage, only use text to balance image realism. However, we found this approach yielded minimal effectiveness. We assumed $\beta = 1$ in the first $t$ denoising steps and $\beta = 0$ in the subsequent denoising steps. As shown in Fig. 1 (b), the object's position is already determined around 20 steps. However, it is common that the generated images exhibit almost no difference between $t = 20$ and $t = 50$. This suggests that even when the injection of layout is stopped in the later denoising stages, the control of text alone still fails to alleviate the poor realism issue. The trade-off between realism and compositionality in T2I and L2I models is challenging yet necessary.

To this end, we introduce a general *training-free* and *transferred-friendly* text-to-image generation framework ***RealCompo***, which utilizes a novel *balancer* to achieve dynamic equilibrium between realism and compositionality in generated images. We first utilize LLMs to generate scene layouts from text prompt through in-context learning [32]. Then we propose an innovative *balancer* to dynamically compose pre-trained fidelity-aware (T2I, stylized T2I) and spatial-aware (e.g., layout, keypoint, segmentation map) image diffusion models. This balancer automatically adjusts the coefficient of the predicted noise for each model by analyzing their cross-attention maps during the denoising stage. By combining the respective strengths of the two models, it achieves a trade-off between realism and compositionality. Finally, we extend RealCompo to various spatial-aware conditions through a general compositional denoising process. Moreover, by changing the T2I model to a stylized T2I model, Realcompo can seamlessly achieve compositional generation specified with a particular style. These dramatically demonstrate the great generalization ability of RealCompo. Although there exist methods [61, 2] for composing multiple diffusion models, their application lacks flexibility because they require additional training and cannot be generalized to other conditionss and models. Our method effectively composes two models in a training-free manner, allowing for a seamless transition between various models.

To the best of our knowledge, RealCompo effectively achieves a trade-off between realism and compositionality in text-to-image generation. Choosing one (stylized) T2I model and one spatial-aware (e.g., layout, keypoint, segmentation map) image diffusion model, RealCompo automatically balances their fidelity and spatial-awareness to realize a collaborative generation. We expands the family of model ensembling/checkpoint merging techniques, which are extensively used in the diffusion community. We believe RealCompo opens up a new research perspective in controllable and compositional image generation.

Our main contributions are summarized as the following:

- We introduce a new *training-free* and *transferred-friendly* text-to-image generation framework RealCompo, which enhances compositional text-to-image generation by balancing the realism and compositionality of generated images.

- We design a novel *balancer* to dynamically combine the predict noise from T2I model and spatial-aware (e.g., layout, keypoint, segmentation map) image diffusion model.

- RealCompo has strong flexibility, can be generalized to balance various (stylized) T2I models and spatial-aware image diffusion models and can achieve high-quality compositional stylized generation. It provides a fresh perspective for compositional image generation.

- Extensive qualitative and quantitative comparisons with previous outstanding methods demonstrate that RealCompo has significantly improved the performance in generating multiple objects and complex relationships.

## 2 Related Work

**Text-to-Image Generation** In recent years, the field of text-to-image generation has made remarkable progress [47, 60, 36, 18, 11, 74, 63], largely attributed to breakthroughs in diffusion models. By training on large-scale image-text paired datasets, T2I models such as Stable Diffusion (SD) [41], DALL-E 2/3 [39, 4], MDM [17], and Pixart-$\alpha$ [7], have demonstrated remarkable generative capabilities. However, there is still significant room for improvement in compositional generation when text prompts include multiple objects and complex relationships [58]. Many studies have attempted to address this issue through controllable generation [72] by providing additional conditions such as segmentation map [22], scene graph [62], layout [77], etc., to constrain the model's generative direction to ensure the accuracy of the number and position of objects in the generated images. However, due to the constraints of the additional conditions, image realism may decrease [27]. Furthermore, several works [37, 9, 68, 65, 30] have attempted to bridge the language understanding gap in models by pre-processing prompts with Large Language Models (LLMs) [1, 48]. It is challenging for T2I models to achieve trade-off between realism and compositionality [65] of generated images.

**Compositional Text-to-Image Generation** Recently, numerous methods have been introduced to improve compositional text-to-image generation [53, 78, 69, 55, 25, 29]. These methods enhance diffusion models in attribute binding, object relationship, numeracy, and complex prompts. Recent studies can generally be divided into two types [52]: one primarily uses cross-attention maps for compositional generation [31, 24, 76], while the other provides more conditions (e.g., layout, keypoint, segmentation map) to achieve controllable generation [16, 78]. The first methods delve into a detailed analysis of cross-attention maps, particularly emphasizing their correspondence with the text prompt. Attend-and-Excite [6] dynamically intervenes in the generation process to improve the model's generation results in terms of attribute binding (such as color). Most of the second methods offer layout as a constraint, enabling the model to generate images that meet this condition. This approach directly defines the area where objects are located, making it more straightforward and observable compared to the first type of methods [27]. LMD [28] provides an additional layout as input with LLMs. Afterward, a controller is designed to predict the masked latent for each object's bounding box and combine them in the denoising process. However, these algorithms are unsatisfactory in the realism of generated images. A recent powerful framework RPG [65] utilizes Multimodal LLMs to decompose complex generation tasks into simpler subtasks to obtain satisfactory realism and compositionality of generated images. Orthogonal to this work, we achieve dynamic equilibrium between realism and compositionality by combining T2I and spatial-aware image diffusion models.

## 3 Method

In this section, we introduce our method, RealCompo, which designs a novel balancer to achieve dynamic equilibrium between realism and compositionality of generated images. We initially focus on the layout-to-image models. In Section 3.1, we analyze the necessity of incorporating influence for the predictive noise of each model and provide a method for calculating coefficients. In Section 3.2, we provide a detailed explanation of the update rules employed by the balancer, which utilizes a training-free approach to update coefficients dynamically. In Section 3.3, we provide a universal formula and denoising procedure that enable the balance of T2I models with any spatial-aware image diffusion model, such as keypoint or segmentation-to-image models based on ControlNet [72]. We also extend RealCompo to stylized compositional generation by stylized T2I models.

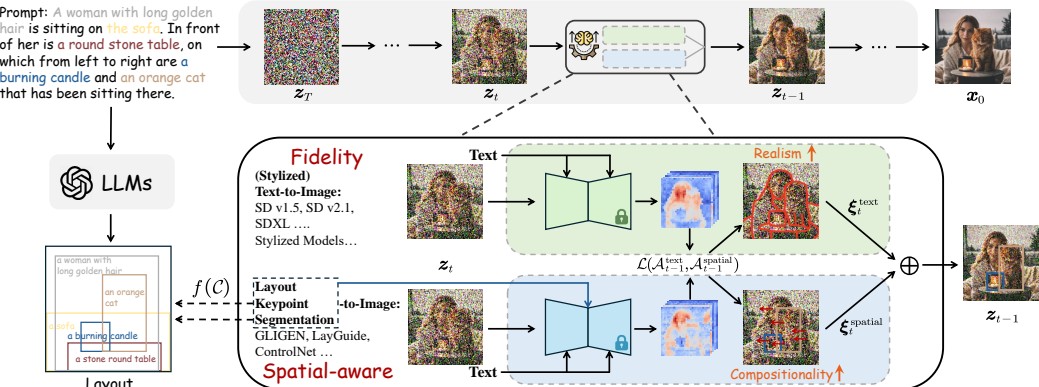

Figure 2: An overview of RealCompo framework for text-to-image generation. We first use LLMs or transfer function to obtain the corresponding layout. Next, the balancer dynamically updates the influence of two models, which enhances realism by focusing on contours and colors in the fidelity branch, and improves compositionality by manipulating object positions in the spatial-aware branch.

## 3.1 Combination of Fidelity and Spatial-Awareness

**LLM-based Layout Generation.**    Since spatial-aware conditions are similar essentially, we first choose layout as the representative of spatial-aware condition for introduction. As shown in Fig. 2, we leverage the powerful in-context learning [57, 79] capability of Large Language Models (LLMs) to analyze the input text prompt and generate an accurate layout to achieve "pre-binding" between objects and attributes. The layout is then used as input for the L2I model. In this paper, we choose GPT-4 for layout generation. Please refer to Appendix C.1 for detailed explanation.

**Combination of Two Types of Noise.**    In diffusion models, the model's predicted noise $\epsilon_t$ directly affects the direction of the generated images. In T2I models, $\epsilon_t^{\text{text}}$ exhibits more directive toward realism [41], whereas in L2I models, $\epsilon_t^{\text{layout}}$ demonstrates more directive toward compositionality [27]. To achieve the trade-off between realism and compositionality, a feasible but untapped solution is to compose the predicted noise of two models. However, the predicted noise from different models has its own generative direction, contributing differently to the generated results at different timesteps and positions. Based on this, we design a novel balancer that achieves dynamic equilibrium between the two models' strengths at every position $i$ in the noise for timestep $t$. This is achieved by analyzing the influence of each model's predicted noise. Specifically, we first set the same coefficient for the predicted noise of each model to represent their influence before the first denoising step:

$$\boldsymbol{Coe}_T^{\text{text}} = \boldsymbol{Coe}_T^{\text{layout}} \tag{1}$$

In order to regularize the influence of each model, we perform a softmax operation on the coefficients to get the final coefficients:

$$\boldsymbol{\xi}_t^c = \frac{\exp(\boldsymbol{Coe}_t^c)}{\exp(\boldsymbol{Coe}_t^{\text{text}}) + \exp(\boldsymbol{Coe}_t^{\text{layout}})} \tag{2}$$

where $c \in \{\text{text}, \text{layout}\}$.

The balanced noise can be derived according to the coefficient of each model:

$$\boldsymbol{\epsilon}_t = \boldsymbol{\xi}_t^{\text{text}} \odot \boldsymbol{\epsilon}_t^{\text{text}} + \boldsymbol{\xi}_t^{\text{layout}} \odot \boldsymbol{\epsilon}_t^{\text{layout}} \tag{3}$$

where $\odot$ denotes pixel-wise multiplication.

Once the predicted noise $\boldsymbol{\epsilon}_t^c$ and the coefficient $\boldsymbol{Coe}_t^c$ of each model are provided, the balanced noise can be derived from Eq. 2 and Eq. 3. At timestep $t$, the balancer dynamically updates coefficients as described in Section 3.2.

## 3.2 Influence Estimation with Dynamic Balancer

The alignment between the generated images and the input prompts is largely influenced by model's cross-attention maps, which encapsulate a wealth of matching information between visual and textual elements, such as location and shape. Specifically, given the intermediate feature $\varphi(\boldsymbol{z}_t)$ and the text embeddings $\tau_\theta(y)$, cross-attention maps can be derived in the following manner:

$$\mathcal{A}^c = \text{Softmax}\left(\frac{Q^c(K^c)^T}{\sqrt{d_k^c}}\right), c \in \{\text{text}, \text{layout}\} \tag{4}$$

$$Q = W_Q \cdot \varphi(\boldsymbol{z}_t), \ K = W_K \cdot \tau_\theta(y) \tag{5}$$

where $Q$ and $K$ are respectively the dot product results of the intermediate feature $\varphi(\boldsymbol{z}_t)$, text embeddings $\tau_\theta(y)$, and two learnable matrices $W_Q$ and $W_K$. $\mathcal{A}_{ij}$ defines the weight of the value of the $j$-th token on the $i$-th pixel. Here, $j \in \{1, 2, \dots, N(\tau_\theta(y))\}$, and $N(\tau_\theta(y))$ denotes the number of tokens in $\tau_\theta(y)$. The dimension of $K$ is represented by $d_k$.

**Update Rule of Dynamic Balancer.** We designed a novel balancer that dynamically balances two models according to their cross-attention maps at timestep $t$. Specifically, we represent layout as $\mathcal{B} = \{b_1, b_2, \dots, b_v\}$, which is composed of $v$ bounding boxes $b$. Each bounding box $b$ corresponds to a binary mask $\mathcal{M}_b$, where the value inside the box is $1$ and the value outside the box is $0$. Given the predicted noise $\epsilon_t^c$ and the coefficient $\boldsymbol{Coe}_t^c$ of each model, the balanced noise $\epsilon_t$ and denoised latent $\boldsymbol{z}_{t-1}$ can be derived from Eq. 3 and Eq. 12. By feeding $\boldsymbol{z}_{t-1}$ into two models, we obtain the cross-attention maps $\mathcal{A}_{t-1}^c$ output by the two models at timestep $t-1$, which indicates the denoising quality feedback after the noise $\epsilon_t^c$ of the model at time $t$ is weighted by $\boldsymbol{\xi}_t^c$. Based on $\mathcal{A}_{t-1}^c$, we define the loss function as follows:

$$\mathcal{L}(\mathcal{A}_{t-1}^{\text{text}}, \mathcal{A}_{t-1}^{\text{layout}}) = \sum_c \sum_b \left(1 - \frac{\sum_i \mathcal{A}_{(ij_b, t-1)}^c \odot \mathcal{M}_b}{\sum_i \mathcal{A}_{(ij_b, t-1)}^c}\right) \tag{6}$$

where $c \in \{\text{text}, \text{layout}\}$, $j_b$ denotes the token corresponding to the object in bounding box $b$. Since two models are controlled by different conditions, averaging the predicted noise equally will lead to instability in the generated images. This is because the T2I model breaks the layout constraints of the L2I model, reducing the compositionality of the generated images, as we have demonstrated in experimrnts in Fig. 9. Therefore, we designed this loss function to measure the alignment between the cross-attention maps and layout for each model. A smaller loss indicates better compositionality. The following rule is used to update $\boldsymbol{Coe}_t^c$:

$$\boldsymbol{Coe}_t^c = \boldsymbol{Coe}_t^c - \rho_t \nabla_{\boldsymbol{Coe}_t^c} \mathcal{L}(\mathcal{A}_{t-1}^{\text{text}}, \mathcal{A}_{t-1}^{\text{layout}}) \tag{7}$$

where $\rho_t$ is the updating rate. This update rule continuously strengthens the constraints on both models by assessing the positional alignment of the layout within the cross-attention maps, ensuring the maintenance of the localization capability of L2I model while injecting fidelity information of T2I model. It is worth noting that previous methods [6, 59, 28] for parameter updates based on function gradients were primarily using energy functions to update latent $\boldsymbol{z}_t$. We are the first to update the influence of predicted noise based on the gradient of the loss function, which is a novel and stable method well-suited to our task. The complete denoising process is detailed in Appendix C.3.

## 3.3 Extend RealCompo to any Spatial-Aware Conditions in a General Form

Other spatial-aware text-to-image diffusion models are essentially similar to L2I models. Keypoint-to-image (K2I) models generate specified actions or poses within each group of keypoints region, and segmentation-to-image (S2I) models fill indicated objects within each segmented region. The concept of "region" is always present, which transforms T2I generation from a macro perspective to utilizing region-based control for T2I generation from a micro perspective. This concept is also the core of enhancing image compositionality. Compared with layout-based T2I generation, the only difference is that keypoints and segmentation maps have stronger control over the model based on regions, requiring that the pose is maintained and the object is correct and unique.

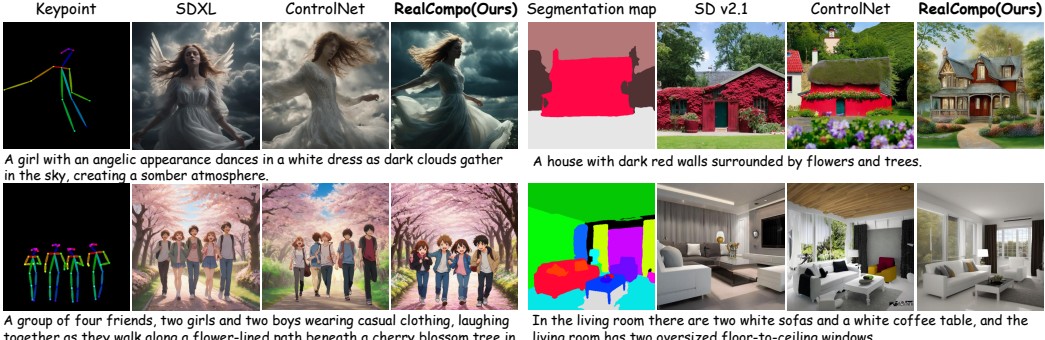

| Keypoint | SDXL | ControlNet | **RealCompo(Ours)** | Segmentation map | SD v2.1 | ControlNet | **RealCompo(Ours)** |

A girl with an angelic appearance dances in a white dress as dark clouds gather in the sky, creating a somber atmosphere.

A house with dark red walls surrounded by flowers and trees.

A group of four friends, two girls and two boys wearing casual clothing, laughing together as they walk along a flower-lined path beneath a cherry blossom tree in full bloom.

In the living room there are two white sofas and a white coffee table, and the living room has two oversized floor-to-ceiling windows.

Figure 3: Extend RealCompo to keypoint- and segmentation-based image generation.

**General Form for Extension to Other Spatial-Aware Conditions** We rethink Eq. 6, which is RealCompo's core approach in combining T2I and L2I models, where the only layout-related variable is the binary masks $\mathcal{M}$. Considering that spatial-aware controllable T2I generation inherently focus on the concept of "region control", we introduce a transfer function:

$$\mathcal{M} = f(\mathcal{C}) \tag{8}$$

where $\mathcal{C}$ represents other spatial-aware conditions such as keypoint and segmentation map. $f(\cdot)$ represents the calculation of the minimum and maximum values of the horizontal and vertical coordinates occupied by each set of keypoints or a segmentation block within the entire image

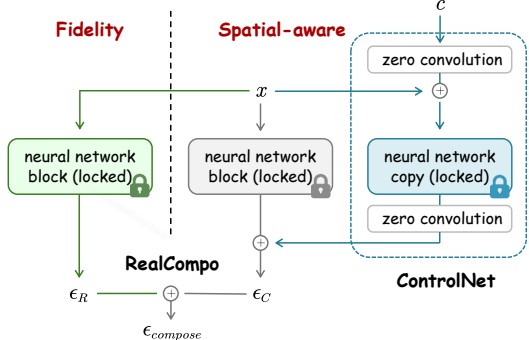

Figure 4: RealCompo constructed on ControlNet.

coordinate system, which can be transformed into a layout and a binary mask $\mathcal{M}$. Therefore, for any T2I models with spatial-aware control, the general loss function of RealCompo is:

$$\mathcal{L}(\mathcal{A}_{t-1}^{\text{text}}, \mathcal{A}_{t-1}^{\text{spatial}}) = \sum_{c} \sum_{b} \left( 1 - \frac{\sum_i \mathcal{A}_{(ij_b, t-1)}^c \odot f_b(\mathcal{C})}{\sum_i \mathcal{A}_{(ij_b, t-1)}^c} \right) \tag{9}$$

where $c \in \{\text{text}, \text{spatial}\}$. Similarly, $\boldsymbol{Coe}_t^c$ is dynamically updated using Eq. 7. ControlNet [72] enables controllable T2I generation based on various spatial-aware conditions. In this work, the spatial-aware branches besides layout are all based on ControlNet, which is illustrated in Fig. 4. The generated images of keypoint- and segmentation-based RealCompo are shown in Fig. 3.

**Extend RealCompo to Stylized Image Generation** As an essential indicator of fidelity, image style [50, 67] guides us to expand the application potential of RealCompo. Since RealCompo mainly leverages T2I models to enhance and guide the realism and aesthetic quality of generated images. By replacing the T2I model with various stylized T2I models and combining it with a spatial-aware image diffusion model, we can achieve outstanding compositional generation under this style. The experiments are shown in Fig 8.

## 4  Experiments

### 4.1  Experimental Setup

**Implementation Details** Our RealCompo is a generic, scalable framework that can achieve the complementary advantages of the model with any chosen (stylized) T2I models and spatial-aware image diffusion models. We selected GPT-4 [1] as the layout generator in our experiments, the detailed rules are described in Appendix C.1. For layout-based RealCompo, we chose SD v1.5 [41] and GLIGEN [27] as the backbone. For keypoint-based RealCompo, we chose SDXL [4] and

Table 1: Evaluation results about compositionality on T2I-CompBench [21]. RealCompo consistently demonstrates the best performance regarding attribute binding, object relationships, numeracy and complex compositions. We denote the best score in blue, and the second-best score in green. The baseline data is quoted from PixArt-$\alpha$ [7].

| Model | Attribute Binding | | | Object Relationship | | Numeracy↑ | Complex↑ |
|---|---|---|---|---|---|---|---|
| | Color ↑ | Shape↑ | Texture↑ | Spatial↑ | Non-Spatial↑ | | |
| Stable Diffusion v1.4 [41] | 0.3765 | 0.3576 | 0.4156 | 0.1246 | 0.3079 | 0.4461 | 0.3080 |
| Stable Diffusion v2 [41] | 0.5065 | 0.4221 | 0.4922 | 0.1342 | 0.3096 | 0.4579 | 0.3386 |
| Structured Diffusion [13] | 0.4990 | 0.4218 | 0.4900 | 0.1386 | 0.3111 | 0.4550 | 0.3355 |
| Attn-Exct v2 [6] | 0.6400 | 0.4517 | 0.5963 | 0.1455 | 0.3109 | 0.4767 | 0.3401 |
| DALL-E 2 [39] | 0.5750 | 0.5464 | 0.6374 | 0.1283 | 0.3043 | 0.4873 | 0.3696 |
| Stable Diffusion XL [4] | 0.6369 | 0.5408 | 0.5637 | 0.2032 | 0.3110 | 0.4988 | 0.4091 |
| PixArt-$\alpha$ [7] | 0.6886 | 0.5582 | 0.7044 | 0.2082 | 0.3179 | 0.5058 | 0.4117 |
| GLIGEN[27] | 0.4288 | 0.3998 | 0.3904 | 0.2632 | 0.3036 | 0.4970 | 0.3420 |
| LMD+[28] | 0.4814 | 0.4865 | 0.5699 | 0.2537 | 0.2828 | 0.5762 | 0.3323 |
| **RealCompo (Ours)** | 0.7741 | 0.6032 | 0.7427 | 0.3173 | 0.3294 | 0.6592 | 0.4657 |

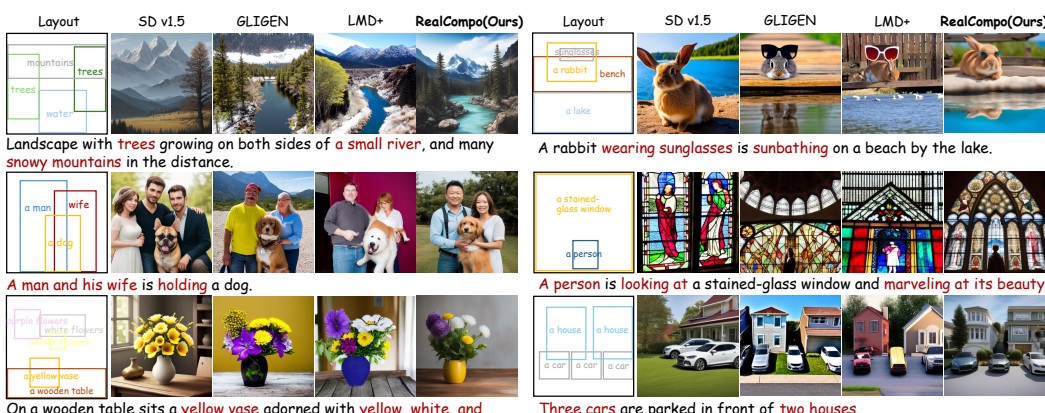

Figure 5: Qualitative comparison between our RealCompo and the outstanding text-to-image model Stable Diffusion v1.5 [41], as well as the layout-to-image models, GLIGEN [27] and LMD+ [28]. Colored text denotes the advantages of RealCompo in generated images.

ControlNet [72] as the backbone. For segmentation-based RealCompo, we chose SD v2.1 [41] and ControlNet [72] as the backbone. For style-based RealCompo, we chose two stylized T2I models: Coloring Page Diffusion and CuteYukiMix as the backbone, and chose GLIGEN [27] as the backbone of L2I model. All of our experiments are conducted under 1 NVIDIA 80G-A100 GPU.

**Baselines and Benchmark**    To evaluate compositionality, we compare our RealCompo with the outstanding T2I and L2I models on T2I-CompBench [21]. This benchmark test models across aspects of attribute binding, object relationship, numeracy and complexity. To evaluate realism, we randomly select 3K text prompts from the COCO validation set , we utilize ViT-B-32 [10] to calculate the CLIP score and LAION aesthetic predictor to calculate aesthetic score, reflecting the degree of match between generated images and prompts as well as the aesthetic quality, respectively. In addition to objective evaluations, we conducted a user study to evaluate RealCompo and stylized RealCompo in terms of realism, compositionality, and comprehensive evaluation.

## 4.2   Main Results

**Results of Compositionality: T2I-CompBench**    We conducted tests on T2I-CompBench [21] to evaluate the compositionality of RealCompo compared to the outstanding T2I and L2I models. As demonstrated in Table 1, RealCompo achieved state-of-the-art performance on all seven evaluation tasks. It is clear that RealCompo and L2I models GLIGEN [27] and LMD+ [28] show significant improvements in spatial-aware tasks such as spatial and numeracy. These improvements are largely

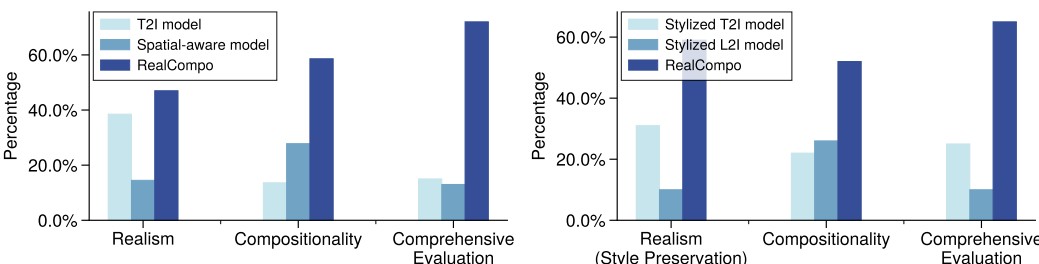

Figure 6: Results of user study.

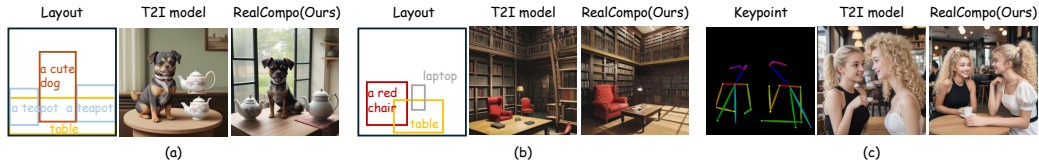

Figure 7: Text-to-image models often generate unrealistic images due to unreasonable object positions. Our method improves image authenticity through conditional control while maintaining detail and aesthetic quality.

attributed to the guidance provided by the additional conditions, which greatly enhances the model's compositional performance. RealCompo employs a balancer for better control over positioning, boosting its advantages in these aspects. However, the L2I models exhibit a noticeable decline in performance on tasks like texture and non-spatial. This decline is due to the injection of layout embeddings, which dilute the density of text embeddings, leading to suboptimal semantic understanding by the model. By composing additional T2I models, RealCompo provides sufficient textual information during the denoising process and achieves outstanding results in tasks that reflect realism, such as texture, non-spatial and complex tasks. As shown in Fig. 5, compared with the current outstanding L2I models GLIGEN and LMD+, RealCompo achieves a high level of realism while keeping the attributes of the objects matched and the number of positions generated correctly.

**Results of Realism: Quantitative Comparison** As shown in Table 2, our model significantly outperforms existing outstanding T2I and L2I models in both CLIP score and aesthetic score. We attribute this to the dynamic balancer, which enhances image realism and aesthetic quality while maintaining high compositionality.

Table 2: Evaluation results on image realism.

| Model | CLIP Score↑ | Aesthetic Score↑ |
|---|---|---|
| Stable Diffusion v1.4 [41] | 0.307 | 5.326 |
| TokenCompose v2.1 [54] | 0.323 | 5.067 |
| Stable Diffusion v2.1 [41] | 0.321 | 5.458 |
| Stable Diffusion XL [4] | 0.322 | 5.531 |
| Layout Guidance[8] | 0.294 | 4.947 |
| GLIGEN[27] | 0.301 | 4.892 |
| LMD+[28] | 0.298 | 4.964 |
| **RealCompo (Ours)** | 0.334 | 5.742 |

**User Study** In addition to objective evaluations, we designed a user study to subjectively assess the practical performance of various methods. We randomly selected 15 prompts, including 5 for stylization experiments. Comparative tests were conducted using T2I models, spatial-aware image diffusion models, and RealCompo. We invited 39 users from diverse backgrounds to vote on image realism, image compositionality, and comprehensive evaluation, resulting in a total of 1755 votes. As illustrated in Fig. 6, RealCompo received widespread user approval in terms of realism and compositionality.

**Reasonable Composition Improves Realism** We provide examples from the user study in Fig. 7, which demonstrates the advantages of RealCompo over the T2I model in realism. As shown in Fig. 7(a), T2I model generates a teapot that is visibly suspended in the air, which doesn't conform to the physical laws of real-world scenes. In contrast, RealCompo generates objects within reasonable

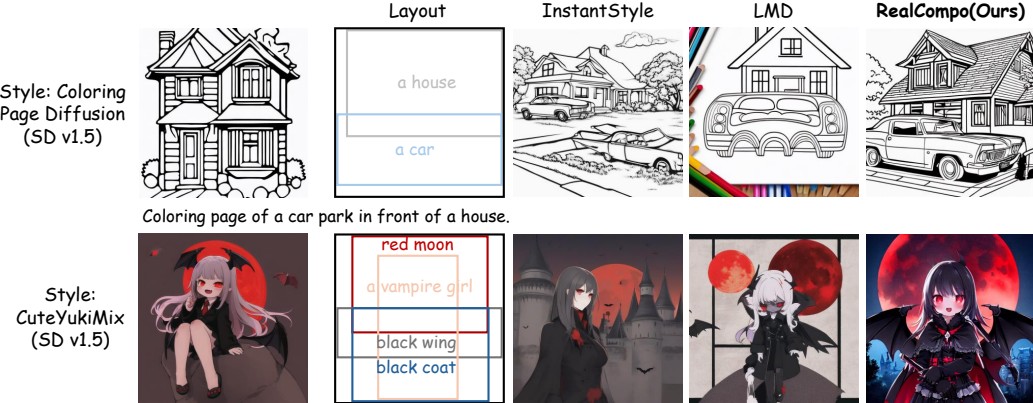

Figure 8: Extend RealCompo to stylized compositional generation.

bounds through layout constraints, ensuring both the aesthetic quality and positional reasonableness. In Fig. 7(b), the red chair generated by the T2I model is unnaturally placed on top of the table, and in Fig. 7(c), two people generated by the T2I model are too close to each other. These examples illustrate that although T2I model outperforms in detail and visual refinement, its positional reasonableness needs improvement. Our method utilizes LLM to generate conditions that comply with physical laws, guiding the model to generate images with both high positional reasonableness and aesthetic quality. Therefore, under similar detail and aesthetic quality, RealCompo's more reasonable composition gives it an advantage over the T2I models in terms of realism.

**Results of Extend Applications: More Spatial-Aware Conditions**  We extend RealCompo to more spatial-aware controlled image generation. As shown in Fig. 3, keypoint- and segmentation-based RealCompo achieves outstanding performance in both realism and compositionality. This promising result reveals that as spatial-aware conditions, layout, keypoint, and segmentation map are fundamentally similar, RealCompo focuses on these similarities and achieves a general generative paradigm for compositional generation.

**Results of Extend Applications: Stylized Generation**  Image style is an essential indicator of fidelity. We experiment with generalizing RealCompo to various pre-trained stylized T2I models. We selected the Coloring Page Diffusion and Cutyukimix as the foundational stylized models, focusing on the coloring page style and adorable style, respectively. As shown in Fig. 8, RealCompo perfectly inherits the style of the T2I models and, with the help of L2I model, achieves powerful compositional generation under these styles, which is currently difficult for stylized diffusion models to accomplish. We found it difficult for LMD to strictly maintain the style by simply replacing the backbone with a stylized model, often leading to text leakage [13]. For example, terms like "crayon" frequently appear in the coloring page style, indicating that the layout control disrupts the style or text control, making it challenging for L2I models to achieve stylized compositional generation. In contrast, by maintaining image realism and style, RealCompo demonstrates strong compositionality while better preserving the style compared to currently outstanding stylized models like InstantStyle [50].

### 4.3 Ablation Study

**Importance of Dynamic Balancer**  As shown in Fig. 9, we conducted experiments on the importance of the dynamic balancer. It is clear that without the use of the dynamic balancer, the generated images do not align with the layout. This is because the predicted noise in T2I model is not constrained by the layout, leading to the model generating the object at any position, and the quantity is uncontrollable. Although the image realism is high, the predicted noise of T2I model disrupts the object distribution of the predicted noise of L2I model, leading to poor compositionality of the generated images and uncontrollable in the generation process.

**Generalizing to Different Backbones**  To explore the generalizability of RealCompo for various models, we choose two T2I models, SD v1.5 [41] and TokenCompose [54], and two L2I models, GLIGEN [27] and LayGuide (Layout Guidance) [8]. We combine them two by two, yielding four

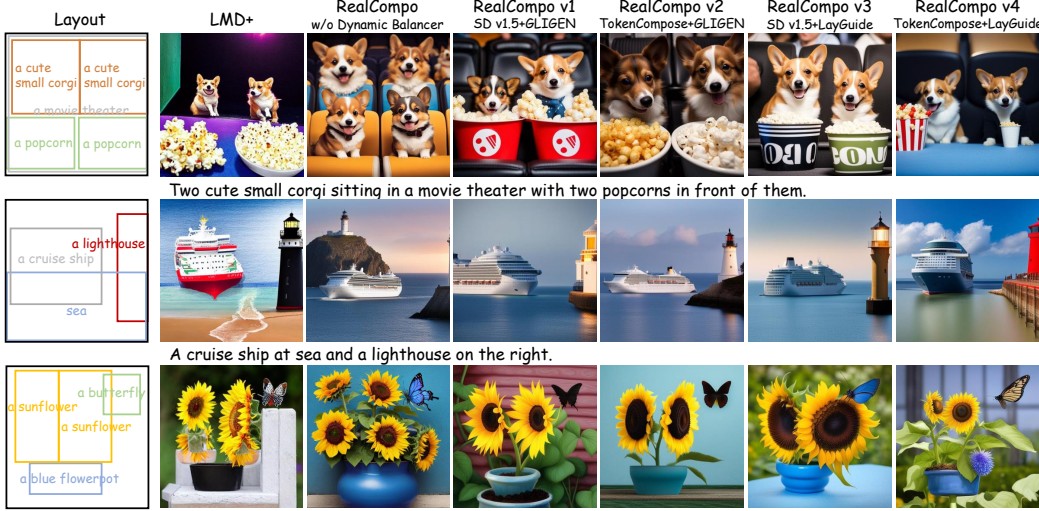

Figure 9: Ablation study on the significance of the dynamic balancer and qualitative comparison of RealCompo's generalization to different models. We demonstrate that dynamic balancer is important to compositional generation and RealCompo has strong generalization and generality to different models, achieving a remarkable level of both fidelity and precision in aligning with text prompts.

versions of RealCompo v1-v4. The experimental results are shown in Fig. 9. The four versions of RealCompo all have a high degree of realism in generating images and achieving desirable results regarding instance composition. This is attributed to the dynamic balancer combining the strengths of T2I and L2I models, and it can seamlessly switch between models because it is simple and requires no training. We also found that RealCompo, when using GLIGEN as the L2I model, performs better than when using LayGuide in generating objects that match the layout. For instance, in the images generated by RealCompo v4 in the first and third rows, "popcorns" and "sunflowers" do not fill up the bounding box, which can be attributed to the superior performance of the base model GLIGEN compared to LayGuide. Therefore, when combined with more powerful T2I and L2I models, RealCompo is expected to yield more satisfactory results.

## 5 Conclusion

In this paper, to solve the challenge of complex or compositional text-to-image generation, we propose the SOTA training-free and transferred-friendly framework RealCompo. In RealCompo, we propose a novel balancer that dynamically combines the advantages of various (stylized) T2I and spatial-aware (e.g., layout, keypoint, segmentation map) image diffusion models to achieve the trade-off between realism and compositionality in generated images. In future work, we will continue to improve this framework by using a more powerful backbone and extend it to more realistic applications.

## Acknowledgement

This work is supported by National Natural Science Foundation of China (U23B2048, U22B2037), Beijing Municipal Science and Technology Project (Z231100010323002), research grant No. SH-2024JK29, Alibaba Cloud, and High-performance Computing Platform of Peking University. This work is also supported by the National Natural Science Foundation of China (Grant No.U1903213) and the Shenzhen Science and Technology Program (JSGG20220831093004008).

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

## A Preliminary

Diffusion models [19, 44, 5] are probabilistic generative models. They can perform multi-step denoising on random noise $x_T \sim \mathcal{N}(0, I)$ to generate clean images through training. Specifically, a gaussian noise $\epsilon$ is gradually added to the clean image $x_0$ in the forward process:

$$x_t = \sqrt{\bar{\alpha}_t} x_0 + \sqrt{1 - \bar{\alpha}_t} \epsilon \tag{10}$$

where $\epsilon \sim \mathcal{N}(0, I)$ and $\alpha_t$ is the noise schedule.

Training is performed by minimizing the squared error loss:

$$\min_{\boldsymbol{\theta}} \mathcal{L} = \mathbb{E}_{x, \epsilon \sim \mathcal{N}(0, I), t} \left[ \| \epsilon - \epsilon_{\boldsymbol{\theta}}(x_t, t) \|_2^2 \right] \tag{11}$$

The parameters of the estimated noise $\epsilon_{\boldsymbol{\theta}}$ are updated step by step by calculating the loss between the real noise $\epsilon$ and the estimated noise $\epsilon_{\boldsymbol{\theta}}(x_t, t)$.

The reverse process aims to start from the noise $x_T$, and denoise it according to the predicted noise $\epsilon_{\boldsymbol{\theta}}(x_t, t)$ at each step. DDIM [45] is a deterministic sampler with denoising steps:

$$x_{t-1} = \sqrt{\bar{\alpha}_{t-1}} \left( \frac{x_t - \sqrt{1 - \bar{\alpha}_t} \epsilon_{\boldsymbol{\theta}}(x_t, t)}{\sqrt{\bar{\alpha}_t}} \right) + \sqrt{1 - \bar{\alpha}_{t-1}} \epsilon_{\boldsymbol{\theta}}(x_t, t) \tag{12}$$

Stable Diffusion [41] is a significant advancement in this field, which conducts noise addition and removal in the latent space. Specifically, SD uses a pre-trained autoencoder that consists of an encoder $\mathcal{E}$ and a decoder $\mathcal{D}$. Given an image $x$, the encoder $\mathcal{E}$ maps $x$ to the latent space, and the decoder $\mathcal{D}$ can reconstruct this image, i.e., $z = \mathcal{E}(x)$, $\tilde{x} = \mathcal{D}(z)$. Moreover, Stable Diffusion supports an additional text prompt $y$ for conditional generation. $y$ is transformed into text embeddings $\tau_\theta(y)$ through the pre-trained CLIP [38] text encoder. $\epsilon_{\boldsymbol{\theta}}$ is trained via:

$$\min_{\boldsymbol{\theta}} \mathcal{L} = \mathbb{E}_{z \sim \mathcal{E}(x), \epsilon \sim \mathcal{N}(0, I), t} \left[ \| \epsilon - \epsilon_{\boldsymbol{\theta}}(z_t, t, \tau_\theta(y)) \|_2^2 \right] \tag{13}$$

In the inference process, noise $z_T \sim \mathcal{N}(0, I)$ is sampled from the latent space. By applying Eq. 12, we perform step-by-step denoising to obtain a clean latent $z_0$. The generative image is then reconstructed through the decoder $\mathcal{D}$.

# B    More Visualized Results on Motivation

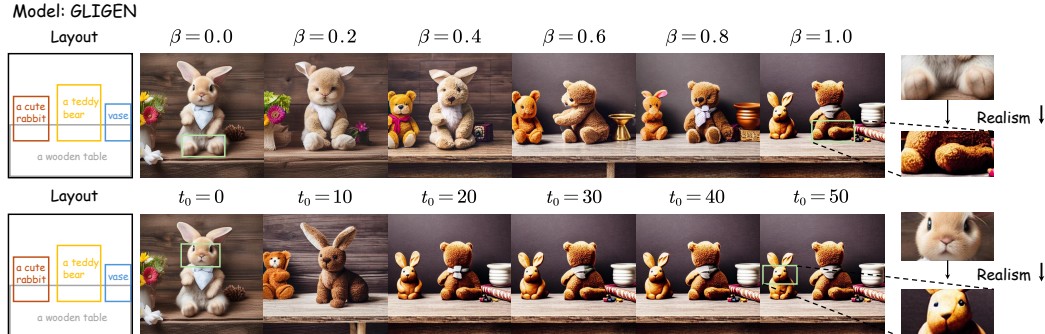

Figure 10: A more intuitive and clearer example to showcase our discoveries and motivation, using GLIGEN [27].

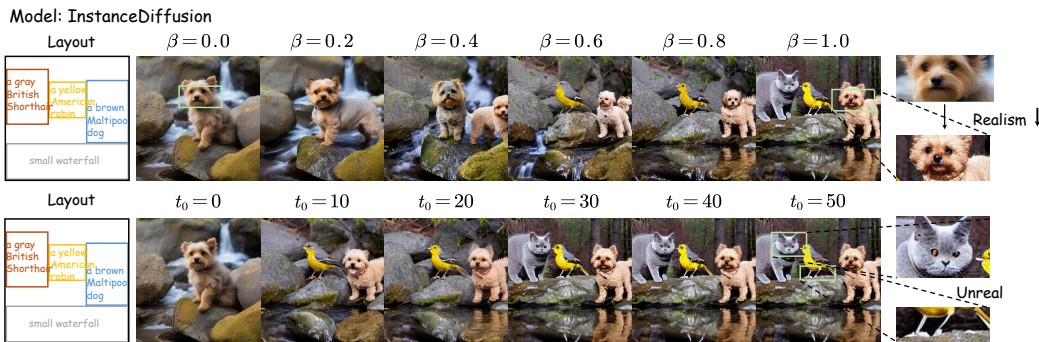

Figure 11: A more intuitive and clearer example to showcase our discoveries and motivation, using InstanceDiffusion [53].

To further verify the generality of the phenomenon we discovered in our motivation. As shown in Fig. 10, we first conducted more experiments on GLIGEN [27]. We observed that as the layout control increased (with a higher $\beta$) or the number of layout control steps increased (with a higher $t_0$), the realism of the generated images declined. There is a noticeable degradation in both detail richness and aesthetic quality. For instance, the legs of the teddy bear appear unrealistic, as if it is facing backward with strange distortions, and the overall details of the rabbit become blurred and unappealing.

Similarly, as shown in Fig. 11, we performed experiments using InstanceDiffusion [53], where we also define a parameter $\beta$ to control the strength of the layout control. It is evident that there is significant quality degradation in the dog's facial and body details. Additionally, the cat's eyes are different sizes, and the bird's legs are abnormally thin, indicating reduced realism in the generated images under the influence of layout control. This suggests that achieving a balance between realism and compositionality in generated images is generally unattainable.

## C    Additional Analysis

### C.1    LLM-based Layout Generation

Large Language Models (LLMs) have witnessed remarkable advancements in recent years [48, 23, 51, 75, 70]. Due to their robust language comprehension, induction, reasoning, and summarization capabilities, LLMs have made significant strides in the Natural Language Processing (NLP) tasks [15, 56]. In the context of multiple-object compositional generation, text-to-image diffusion models

exhibit a relatively weaker understanding of language, as reflected in the poor compositionality of the generated images. Consequently, exploring ways to harness the inferential and imaginative capacities of LLMs to facilitate their collaboration with text-to-image diffusion models, thereby producing images that adhere to the prompt, offers substantial research potential.

In our task, we leverage LLMs to directly infer the layout of all objects based on the user's input prompt through in-context learning (ICL) [26, 43]. This layout is used for the layout-to-image model of RealCompo, eliminating the need to manually provide a layout for each prompt and achieve pre-binding of multiple objects and attributes. Specifically, as shown in Fig. 12, we construct prompt templates, which include descriptions of task rules (instruction), in-context examples (demonstration), and the user's input prompt (test). Through imitation reasoning based on the instruction, LLM generate layout for each object, where each layout represents the coordinates of the top-left and bottom-right corners of a respective box. We selected the highly capable GPT-4 [1] as layout generator.

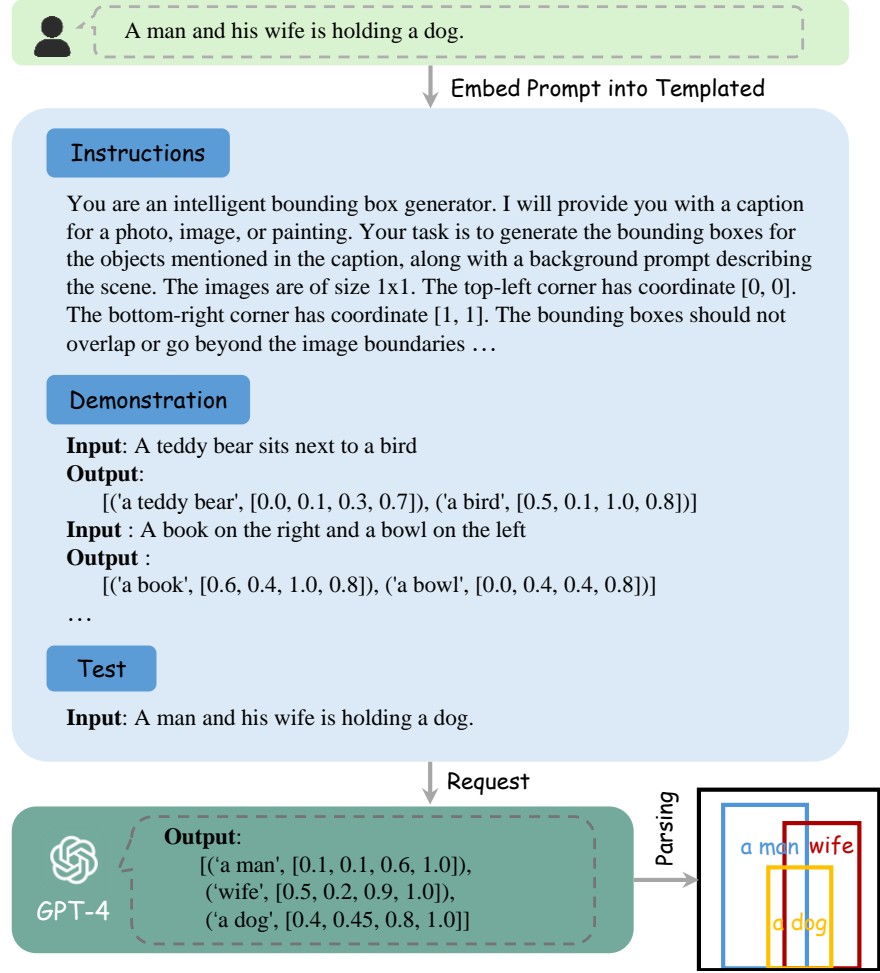

Figure 12: Firstly, the user's input text is embedded into the prompt template. The template is then parsed using GPT-4 with frozen parameters, which yields descriptions of the objects in the prompt as well as their corresponding layout.

## C.2 Analysis of the Existence of Gradient in Eq. 7

Here we set:

$$
\begin{aligned}
\mathcal{L}(\mathcal{A}_{t-1}^{\text{text}}, \mathcal{A}_{t-1}^{\text{layout}}) &= \sum_b \mathcal{L}_b(\mathcal{A}_{t-1}^{\text{text}}, \mathcal{A}_{t-1}^{\text{layout}}) \\
&= \sum_b \left[ \left( 1 - \frac{\sum_i \mathcal{A}_{(ij_b,t-1)}^{\text{text}} \odot \mathcal{M}_b}{\sum_i \mathcal{A}_{(ij_b,t-1)}^{\text{text}}} \right) + \left( 1 - \frac{\sum_i \mathcal{A}_{(ij_b,t-1)}^{\text{layout}} \odot \mathcal{M}_b}{\sum_i \mathcal{A}_{(ij_b,t-1)}^{\text{layout}}} \right) \right]
\end{aligned}
\tag{14}
$$

If the loss function is given by Eq. 6, the gradient in Eq. 7 can be derived as follows:

$$
\begin{aligned}
&\frac{\partial \mathcal{L}\left(\mathcal{A}_{t-1}^{\text{text}}, \mathcal{A}_{t-1}^{\text{layout}}\right)}{\partial \boldsymbol{Coe}_t^c} \\
&= \frac{\partial \sum_b \mathcal{L}_b\left(\mathcal{A}_{t-1}^{\text{text}}, \mathcal{A}_{t-1}^{\text{layout}}\right)}{\partial \boldsymbol{Coe}_t^c} \\
&= \sum_b \frac{\partial \mathcal{L}_b\left(\mathcal{A}_{t-1}^{\text{text}}, \mathcal{A}_{t-1}^{\text{layout}}\right)}{\partial \boldsymbol{Coe}_t^c} \\
&= \sum_b \left[ \frac{\partial \mathcal{L}_b\left(\mathcal{A}_{t-1}^{\text{text}}, \mathcal{A}_{t-1}^{\text{layout}}\right)}{\partial \mathcal{A}_{(j_b,t-1)}^c} \frac{\partial \mathcal{A}_{(j_b,t-1)}^c}{\partial \boldsymbol{z}_{t-1}} \frac{\partial \boldsymbol{z}_{t-1}}{\partial \boldsymbol{\epsilon}_t} \frac{\partial \boldsymbol{\epsilon}_t}{\partial \boldsymbol{\xi}_t^c} \frac{\partial \boldsymbol{\xi}_t^c}{\partial \boldsymbol{Coe}_t^c} \right] \\
&= \sum_b \left[ \frac{\partial \mathcal{L}_b\left(\mathcal{A}_{t-1}^{\text{text}}, \mathcal{A}_{t-1}^{\text{layout}}\right)}{\partial \mathcal{A}_{(j_b,t-1)}^c} \frac{\partial \mathcal{A}_{(j_b,t-1)}^c}{\partial \boldsymbol{z}_{t-1}} \frac{\partial \boldsymbol{z}_{t-1}}{\partial \boldsymbol{\epsilon}_t} \frac{\partial \boldsymbol{\epsilon}_t}{\partial \boldsymbol{\xi}_t^c} \frac{\exp\left(\boldsymbol{Coe}_t^{\text{text}} + \boldsymbol{Coe}_t^{\text{layout}}\right)}{\left(\exp\left(\boldsymbol{Coe}_t^{\text{text}}\right) + \exp\left(\boldsymbol{Coe}_t^{\text{layout}}\right)\right)^2} \right] \\
&= \sum_b \left[ \frac{\partial \mathcal{L}_b\left(\mathcal{A}_{t-1}^{\text{text}}, \mathcal{A}_{t-1}^{\text{layout}}\right)}{\partial \mathcal{A}_{(j_b,t-1)}^c} \frac{\partial \mathcal{A}_{(j_b,t-1)}^c}{\partial \boldsymbol{z}_{t-1}} \frac{\partial \boldsymbol{z}_{t-1}}{\partial \boldsymbol{\epsilon}_t} \frac{\boldsymbol{\epsilon}_t^c \cdot \exp\left(\boldsymbol{Coe}_t^{\text{text}} + \boldsymbol{Coe}_t^{\text{layout}}\right)}{\left(\exp\left(\boldsymbol{Coe}_t^{\text{text}}\right) + \exp\left(\boldsymbol{Coe}_t^{\text{layout}}\right)\right)^2} \right] \\
&= \sum_b \left[ \frac{\partial \mathcal{L}_b\left(\mathcal{A}_{t-1}^{\text{text}}, \mathcal{A}_{t-1}^{\text{layout}}\right)}{\partial \mathcal{A}_{(j_b,t-1)}^c} \frac{\partial \mathcal{A}_{(j_b,t-1)}^c}{\partial \boldsymbol{z}_{t-1}} \left( \sqrt{1 - \bar{\alpha}_{t-1} - \sigma^2} - \frac{\sqrt{1 - \bar{\alpha}_t}}{\sqrt{\alpha_t}} \right) \right. \\
&\quad \left. \times \frac{\boldsymbol{\epsilon}_t^c \cdot \exp\left(\boldsymbol{Coe}_t^{\text{text}} + \boldsymbol{Coe}_t^{\text{layout}}\right)}{\left(\exp\left(\boldsymbol{Coe}_t^{\text{text}}\right) + \exp\left(\boldsymbol{Coe}_t^{\text{layout}}\right)\right)^2} \right]
\end{aligned}
\tag{15}
$$

For any T2I and L2I models, we have the following:

$$
\frac{\partial \mathcal{L}_b\left(\mathcal{A}_{t-1}^{\text{text}}, \mathcal{A}_{t-1}^{\text{layout}}\right)}{\partial \mathcal{A}_{(j_b,t-1)}^c} = \frac{\mathcal{J} \sum_i \left(\mathcal{A}_{(ij_b,t-1)}^c \odot \mathcal{M}_b\right) - \mathcal{M}_b \sum_i \mathcal{A}_{(ij_b,t-1)}^c}{\left(\sum_i \mathcal{A}_{(ij_b,t-1)}^c\right)^2}
\tag{16}
$$

where $\mathcal{J}$ is a matrix with all elements equal to 1. All variables in Eq. 15 are known, indicating the existence of the gradient in Eq. 7.

When using the loss function given by Eq. 9 under any spatial-aware conditions, the gradient in Eq. 7 can be derived as follows:

$$
\frac{\partial \mathcal{L}\left(\mathcal{A}_{t-1}^{\text{text}}, \mathcal{A}_{t-1}^{\text{spatial}}\right)}{\partial \boldsymbol{Coe}_t^c}
$$

$$
= \sum_b \left[ \frac{\partial \mathcal{L}_b\left(\mathcal{A}_{t-1}^{\text{text}}, \mathcal{A}_{t-1}^{\text{spatial}}\right)}{\partial \mathcal{A}_{(j_b,t-1)}^c} \frac{\partial \mathcal{A}_{(j_b,t-1)}^c}{\partial \boldsymbol{z}_{t-1}} \frac{\partial \boldsymbol{z}_{t-1}}{\partial \boldsymbol{\epsilon}_t} \frac{\partial \boldsymbol{\epsilon}_t}{\partial \boldsymbol{\xi}_t^c} \frac{\partial \boldsymbol{\xi}_t^c}{\partial \boldsymbol{Coe}_t^c} \right]
$$

$$
= \sum_b \left[ \frac{\partial \mathcal{L}_b\left(\mathcal{A}_{t-1}^{\text{text}}, \mathcal{A}_{t-1}^{\text{spatial}}\right)}{\partial \mathcal{A}_{(j_b,t-1)}^c} \frac{\partial \mathcal{A}_{(j_b,t-1)}^c}{\partial \boldsymbol{z}_{t-1}} \left( \sqrt{1 - \bar{\alpha}_{t-1} - \sigma^2} - \frac{\sqrt{1 - \bar{\alpha}_t}}{\sqrt{\alpha_t}} \right) \right.
$$

$$
\left. \times \frac{\boldsymbol{\epsilon}_t^c \cdot \exp\left(\boldsymbol{Coe}_t^{\text{text}} + \boldsymbol{Coe}_t^{\text{spatial}}\right)}{\left(\exp\left(\boldsymbol{Coe}_t^{\text{text}}\right) + \exp\left(\boldsymbol{Coe}_t^{\text{spatial}}\right)\right)^2} \right] \tag{17}
$$

$$
\frac{\partial \mathcal{L}_b\left(\mathcal{A}_{t-1}^{\text{text}}, \mathcal{A}_{t-1}^{\text{spatial}}\right)}{\partial \mathcal{A}_{(j_b,t-1)}^c} = \frac{\mathcal{J} \sum_i \left( \mathcal{A}_{(ij_b,t-1)}^c \odot f_b(\mathcal{C}) \right) - f_b(\mathcal{C}) \sum_i \mathcal{A}_{(ij_b,t-1)}^c}{\left( \sum_i \mathcal{A}_{(ij_b,t-1)}^c \right)^2} \tag{18}
$$

where $c \in \{\text{text}, \text{spatial}\}$.

Therefore, the gradient in Eq. 7 exists for the selection of different loss functions.

### C.3 Inference details

We provide a detailed compositional denoising process for RealCompo, which achieves a complementary balance between the advantages of the T2I model and the spatial-aware diffusion model by combining their predicted noise during the denoising stage. We provide the pseudocode for the compositional denoising process of the layout-based RealCompo as followed, we have highlighted the innovations of our method in blue.

---

**Algorithm 1** Compositional denoising procedure of layout-based RealCompo

---

    **Input:** A text prompt $\mathcal{P}$, a set of layout $\mathcal{B}$, a pretrained T2I model and a pretrained L2I model
    **Output:** A clear latent $\boldsymbol{z}_0$
1:  $\boldsymbol{z}_T \sim \mathcal{N}(\boldsymbol{0}, \mathbf{I})$
2:  $\boldsymbol{Coe}_T^{\text{text}} = \boldsymbol{Coe}_T^{\text{layout}} \sim \mathcal{N}(\boldsymbol{0}, \mathbf{I})$
3:  **for** $t = T, \ldots, 1$ **do**
4:     **if** $t > t_0$ **then**
5:         $\boldsymbol{\epsilon}_t, \_ = \text{L2I}\left(\boldsymbol{z}_t, \mathcal{P}, \mathcal{B}, t\right)$
6:     **else**
7:         $\boldsymbol{\epsilon}_t^{\text{text}}, \_ = \text{T2I}\left(\boldsymbol{z}_t, \mathcal{P}, t\right)$
8:         $\boldsymbol{\epsilon}_t^{\text{layout}}, \_ = \text{L2I}\left(\boldsymbol{z}_t, \mathcal{P}, \mathcal{B}, t\right)$
9:         Get the balanced noise $\boldsymbol{\epsilon}_t$ from Eq. 2 and Eq. 3
10:      Get the denoised latent $\boldsymbol{z}_{t-1}$ from Eq. 12
11:      $\boldsymbol{\epsilon}_{t-1}^{\text{text}}, \mathcal{A}_{t-1}^{\text{text}} = \text{T2I}\left(\boldsymbol{z}_{t-1}, \mathcal{P}, t\right)$
12:      $\boldsymbol{\epsilon}_{t-1}^{\text{layout}}, \mathcal{A}_{t-1}^{\text{layout}} = \text{L2I}\left(\boldsymbol{z}_{t-1}, \mathcal{P}, \mathcal{B}, t\right)$
13:      Compute $\mathcal{L}(\mathcal{A}_{t-1}^{\text{text}}, \mathcal{A}_{t-1}^{\text{layout}})$ from Eq. 6
14:      Update $\boldsymbol{Coe}_t^c$ according to Eq. 7
15:      Get the balanced noise $\boldsymbol{\epsilon}_t$ from Eq. 2 and Eq. 3
16:     **end if**
17:    Get the denoised latent $\boldsymbol{z}_{t-1}$ from Eq. 12
18: **end for**
19: **return** $\boldsymbol{z}_0$

---

## C.4 Gradient Analysis

**Gradient Analysis** We selected RealCompo v3 and v4 to analyze the gradient changes in Eq. 7 across all denoising stages. As shown in Fig. 13, we use the same prompt and random seed to visualize the gradient magnitude changes corresponding to T2I and L2I for each model version. We observe that the gradient magnitude change of RealCompo v4 fluctuated more in the early denoising stages. We argue that TokenCompose, which enhances the composition capability of multiple-object generation by fine-tuning the model using segmentation masks, may overlap in functionality with the layout-based multiple-object generation, and TokenCompose's positioning of objects may not consistently align with the bounding box. Therefore, RealCompo must focus on balancing the positioning of TokenCompose and layout in the early denoising stages, leading to less stable gradients compared to RealCompo v3. Additionally, due to LayGuide's weaker positioning ability compared to GLIGEN, RealCompo v4 may occasionally generate objects with less coverage of the bounding box, as mentioned in the ablation experiment in Section 4.3.

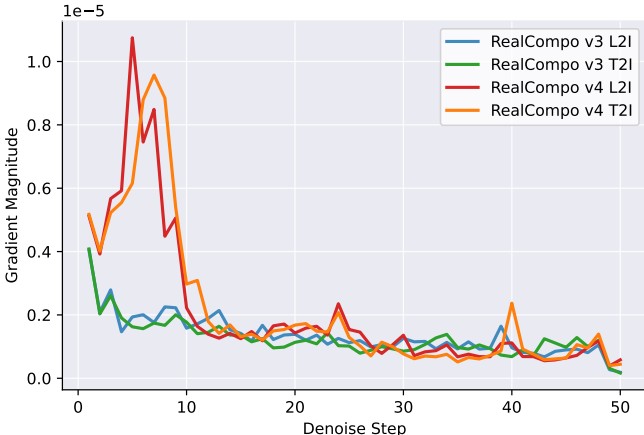

Figure 13: Changes of gradient magnitude in Eq. 7 across all denoising process for the T2I and L2I models of RealCompo v3 and v4.

## C.5 Limitations and Future Work

**Limitations** While our RealCompo enhances both realism and compositionality in a training-free manner, it should be noted that the computational cost of our method is slightly higher compared to that of a single T2I model or a single spatial-aware image diffusion model, due to the need to combine two models and compute loss and gradients. However, by adjusting the combination stage of RealCompo, we can keep the computational cost within an acceptable range.

**Future Work** In future work, we aim to explore more efficient computational methods to improve the calculation efficiency of RealCompo while maintaining high-quality results and we plan to extend its application to more challenging tasks such as text-to-video and text-to-3D generation. Furthermore, given that the exceptional classifier-free guidance strategy employs fixed weights, we aim to explore the potential of using fixed coefficients to further enhance the capabilities of RelCompo.

## C.6 Broader Impact

Recent significant advancements in text-to-image diffusion models have opened up new possibilities for creative design, autonomous media, and various other sectors. However, the dual-use nature of this technology raises concerns about its social impact. Image diffusion models carry the risk of misuse, particularly in the realm of impersonating humans. For example, in today's society, malicious applications such as "deepfakes" have been employed in inappropriate contexts to fabricate attacks on specific public figures. It is crucial to clarify that our algorithm is designed to enhance the quality of image generation, and we do not endorse or facilitate such malicious applications.

# D   More Generation Results

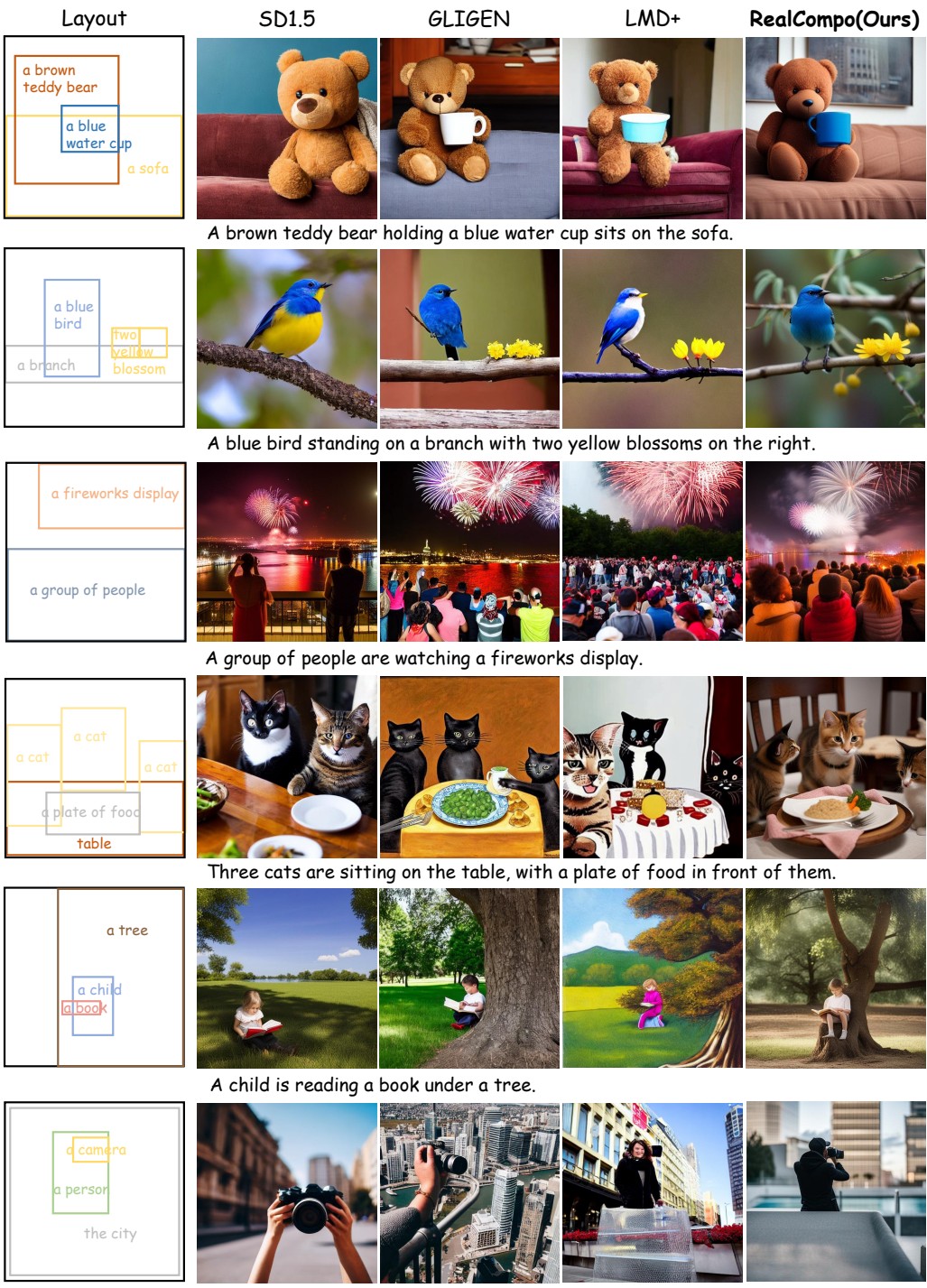

Figure 14: More generation results about layout-based RealCompo.

| Keypoint | SDXL | ControlNet | **RealCompo(Ours)** |
|---|---|---|---|

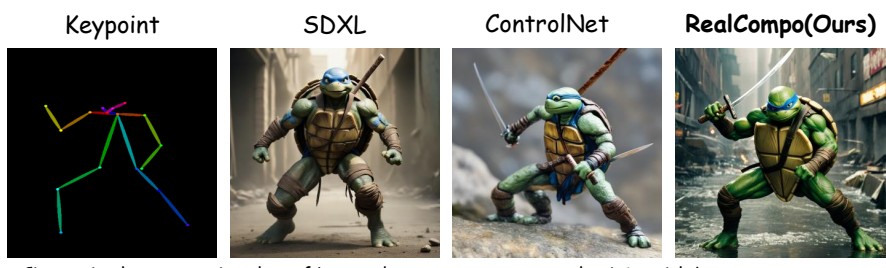

Cinematic photo an action shot of Leonardo teenage mutant turtle ninja, with katana weapon, wet and dirty background

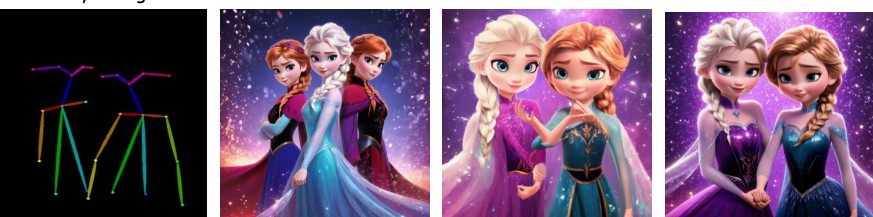

2 girl, Elsa and Anna, sparks of magic between them, princess dress, background with sparkles, black purple red color schemes.

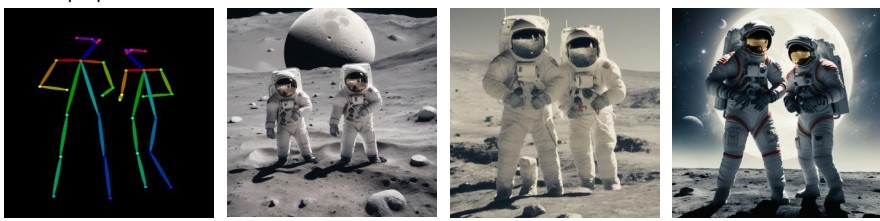

Two astronauts standing on the moon, behind them is a white planet amidst the vast universe.

Figure 15: More generation results about keypoint-based RealCompo.

| Segmentation map | SD v2.1 | ControlNet | **RealCompo(Ours)** |
|---|---|---|---|

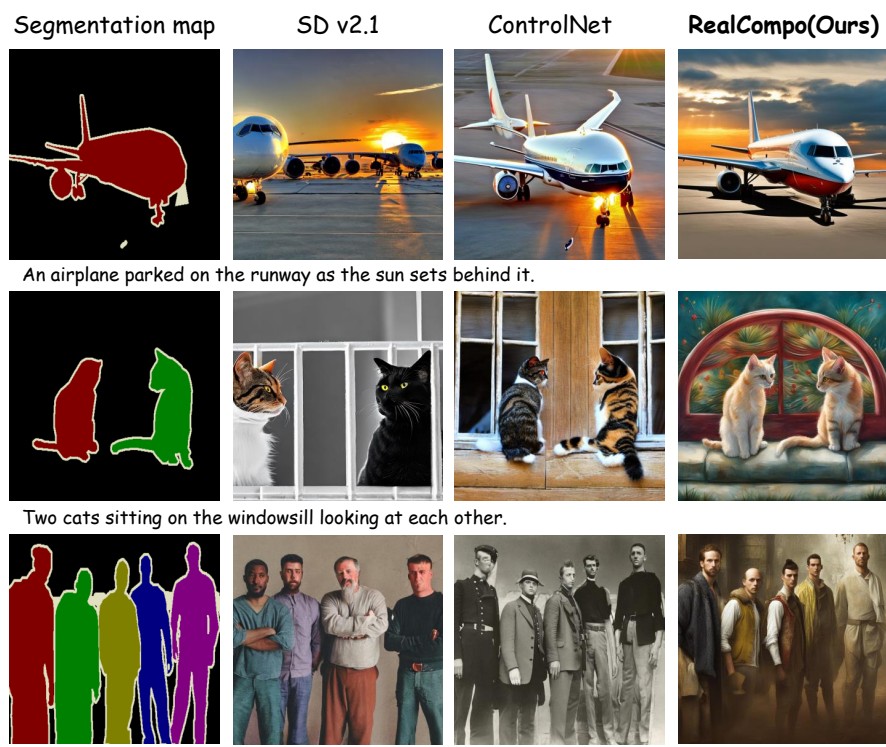

An airplane parked on the runway as the sun sets behind it.

Two cats sitting on the windowsill looking at each other.

Five men stand together in a line, serious in expression.

Figure 16: More generation results about segmentation-based RealCompo.

