# OpenReview forum: "RealCompo: Balancing Realism and Compositionality Improves Text-to-Image Diffusion Models"
_NeurIPS.cc/2024/Conference — NeurIPS 2024 poster_

### Official Review · Reviewer_ZHfz · 2024-06-13

**Soundness:** 3
**Presentation:** 2
**Contribution:** 1
**Rating:** 6
**Confidence:** 5

**Summary:**

This paper proposes to balance the realism and compositionality of the generated images by means of different diffusion models, such as pre-trained T2I models and models based on spatial perceptual control. The authors develop a balancer that optimizes the two coefficients through mask-based attention manipulation to enable dynamic combination of the different models. They also demonstrate the effectiveness of the method through extensive experiments.

**Strengths:**

Pros:
1. The proposed design could be used to bridge different models and  support better image generation.
2. SOTA quantitative and qualitative results obtained by the proposed method.
3. The method can be extended to stylized image generation in a training-free manner.

**Weaknesses:**

Cons:
1. Unclear Task definition. First, even different pre-trained T2I models can result in different so-called REALISM due to different training strategies and data. Secondly, Control-based models, e.g., Controlnet and GLIGEN, all add controllability while inheriting the realism generated by the original T2I models. So I'm not sure why the authors need to propose another method to do the so-called balancing of realism and composability.In addition, how can you guarantee that the combination of T2I and L2I can generate real diagrams when the images generated by T2I are all more fake?
2. Unconvincing observation. In Figure 1 and paragraphs 2 to 4, the authors present their motivation only through an existing method, GLIGEN. I argue that this is inappropriate. After all, there are many methods based on layout or control, no matter what conditions these methods use, such as layout, segmentation map, and pose. Motivation is the key to a paper and cannot be deduced from just one method. So I argue that this observation is not very convincing to me.
3. Insufficient contribution. It is common to use optimization to produce better images, but the difference lies in the design of the optimization. The use of mask to combine with attention to do optimization is more common way, such as in [1,2,3,4,5]. The only difference is that the existing methods may optimize zt, this paper optimizes two coefficients. So I argue that the contribution of this paper is not very sufficient.

**Questions:**

1. Fig-1 lacks a detailed explanation.
2. Sub-sec 3.3 contains not much valuable information, as the only difference between Eq.9 and Eq.6 is how to obtain a mask/spatial condition.

I will revise my rating according to the author's feedback and the reviewer's discussion.

---

> ### Author Rebuttal · Authors · 2024-08-06
>
> *We sincerely thank you for your time and efforts in reviewing our paper, and your valuable feekback. We are glad to see that the proposed method can be generalized to different models, achieve SOTA generation results, and expand to various application scenarios in a training-free manner. Please see below for our responses to your comments.*
>
> **Q1: Why propose another method for balancing realism and composability? T2I models differ in realism due to training variations, and control-based models enhance controllability while retaining original realism.**
>
> A1: Thank you for your comment. As you mentioned, control-based models enhance controllability while retaining the realism of original T2I models. However, **our extensive experiments show that increasing the strength of condition control significantly reduces detail richness and overall aesthetics.** Fig1 in our paper and Figs3 and 4 in supplementary PDF illustrate this decline of realism with increased control strength ($\beta$ increase). Additionally, Fig1 and 2 in supplementary PDF demonstrate that even with larger, more powerful backbones, increasing control strength degrades detail and aesthetic quality. **Thus, control-based models must sacrifice realism to improve compositionality**. The imbalance between realism and compositionality in control-based models is a critical and unresolved issue.
>
> It's correct that different pre-trained T2I models produce varying levels of realism due to different training strategies and data. Hence, **by using different stylized T2I models, our method easily achieves stylized compositional generation**. As shown in Fig7 in our paper, our method maintains high-quality style preservation with different stylized T2I models.
>
> **Q2：How can you ensure that combining T2I and L2I generates real diagrams when T2I images are mostly fake?**
>
> A2: **One important criterion for determining whether an image appears real or fake is the reasonableness of its composition, specifically whether object placement aligns with real-world physical scenarios**. Fig5 of supplementary PDF shows examples where T2I-generated images fail in this regard.  For instance, in Fig5(a), the teapot generated by T2I model is visibly suspended in the air, defying physical laws. Our method uses layout constraints to ensure objects are generated within reasonable bounds, maintaining both aesthetic quality and compositional reasonableness. Similarly, the image generated by T2I model in Fig5(b) shows a red chair unnaturally placed on a table, and in Fig5(c) depicts two people too close to each other. **These examples indicate that while T2I model excels in aesthetics, it lacks compositional reasonableness. Our method uses LLM to generate conditions that comply with physical laws, guiding the model to generate images with high compositional and aesthetic quality.** Thus, our method generates more realistic images compared to T2I models.
>
> **Q3: In Fig1 and paragraphs 2-4, the authors present their motivation solely through GLIGEN, which is inappropriate.**
>
> A3: We are grateful for your feedback and apologize for our oversight. In supplementary PDF, we provided additional experiments to validate our motivation. Fig3 shows a more apparent trend of changes in image realism using GLIGEN. Fig4 uses InstanceDiffusion to further validate this, with a parameter $\beta$ to control the strength of layout guidance. As $\beta$ increases, the realism of generated images decreases, evidenced by loss of details, decline in aesthetic quality, and unrealistic content. For example, the dog's facial and body details degrade, the cat's eyes differ in size, and the bird has abnormally thin legs.
>
> Additionally, we set the parameter $t_0$ to control the number of steps for layout control during generation. Specifically, when denoising steps are fewer than $t_0$, layout is used for control, and when the steps exceed $t_0$, only text is used. Similar to GLIGEN, when $t_0$ exceeds 20, the generated images of InstanceDiffusion show minimal differences, indicating that layout control's influence is mainly in the early denoising stages. Even if layout guidance is discarded in the later stages of denoising and only text is used, it is still challenging to recover the realism of the images.
>
> Fig1 and 2 in supplementary PDF show our analysis of the aesthetic quality of images generated by GLIGEN and ControlNet at different sizes. The figures demonstrate that as conditional control strength increases, image aesthetic quality declines at any size. Thank you again for your feedback. We will expand on this part of the experiments in the manuscript.
>
> **Q4:  Existing methods may optimize $z_t$, while this paper optimizes two coefficients. Thus, the contribution is insufficient.**
>
> A4: Here we would like to emphasize two points that distinguish us from previous work and add to the significance of our findings in this paper, **we provide detailed clarifications in the General Response**.
>
> First, we are the first to discover the imbalance between realism and compositionality in generated images, providing experimental analysis and conclusions.
>
> Second, we offer a new perspective on optimization-based generation methods, achieving satisfactory results and generalization using coefficient updates for the first time.
>
> We will clarify these two points further in our manuscript.
>
> **Q5: Fig1 lacks detailed explanation.**
>
> A5: Thank you for your feedback, and we apologize for any confusion caused. Details about Fig1 can be found in **A3**. We will update our manuscript to provide more comprehensive and accessible explanations regarding Fig1.
>
> **Q6: Sub-sec 3.3 contains limited valuable information since difference between Eq9 and Eq6 is how to obtain a mask/spatial condition.**
>
> A6: Thank you for your suggestion. Sub-sec 3.3 is an extended part of our paper, aimed at explaining the generalizability of spatial-aware conditions. We will condense and refine this section in future revisions.

---

> > ### Author Response · Authors · 2024-08-11
> > **A minor clarify**
> >
> > Dear Esteemed Reviewer ZHfz,
> >
> > We would like to clarify that the supplement PDF refers to **the pdf in global response**. We apologize for any confusion this may have caused.
> >
> > Should there be any further points that require clarification or improvement, please know that we are fully committed to addressing them promptly.
> >
> > Thank you once again for your invaluable contribution to our research.
> >
> > Warm Regards,
> >
> > The Authors

---

> > ### Comment · Reviewer_ZHfz · 2024-08-13
> >
> > My questions are all well addressed, and I decided to increase my rating to weak accept. BTW, PLEASE make the necessary revisions in the final version according to the questions listed above.

---

> > > ### Author Response · Authors · 2024-08-13
> > > **Thank you for your support**
> > >
> > > Thank you very much for raising the score! We sincerely appreciate your valuable comments and the time and effort you put into reviewing our paper.
> > >
> > > We will make sure to incorporate these suggestions into the final manuscript.
> > >
> > > Warm Regards,
> > >
> > > The Authors

---

> ### Author Response · Authors · 2024-08-12
> **Gentle Reminder**
>
> Hello, Reviewer ZHfz, Regarding your concerns, we have provided some responses that you might find useful, including:
>
> - We have highlighted and summarized the innovations and contributions of our method from two perspectives: innovation in task establishment and model design.
>
> - Additional experimental results, incuding using different models to validate our motivation and compare our method with T2I models in terms of the visual authenticity of the images.
>
> - We provide a detailed explanation of Figure 1 in the paper and the motivation of our method.
>
> We sincerely hope our responses address some of your concerns. If you have any further questions, please feel free to ask. Thank you.

---

### Official Review · Reviewer_juPc · 2024-07-11

**Soundness:** 2
**Presentation:** 3
**Contribution:** 3
**Rating:** 6
**Confidence:** 4

**Summary:**

This paper presents a method, named RealCompo, for combining multiple diffusion models, such as a text-to-image model and a spatial-aware one, to achieve the best of both worlds: superior image realism and compositionality. It merges predicted noise from both models during each denoising step, and balances them based on two sets of coefficients that are dynamically updated through a loss that forces cross attention maps to follow provided spatial constraints. Experiments show that RealCompo improves both realism and compositionality compared to using either model alone, and it outperforms SOTA spatial-aware models in compositional generation.

**Strengths:**

- The paper is well written and easy to follow.
- The proposed method is straightforward and compatible across various models.
- The idea of fusing multiple diffusion models during denoising is interesting. This may have applications beyond compositional text-to-image generation.
- Extensive experiments are conducted on various T2I models, spatial-aware models, stylized models.

**Weaknesses:**

- Since the proposed method runs multiple models concurrently and requires gradient-based updates, a discussion on computational efficiency is crucial for its practical application.
- Figure 1 does not illustrate the trade-off between realism and compositionality well. I don't see an apparent drop of realism/aesthetics as $\beta$ (control of layout) increases.

**Questions:**

- Why can RealCompo improve image realism (Table 2, Figure 6 left) and style preservation (Figure 6 right) beyond the capabilities of the T2I model alone? This is difficult to understand because the loss function (Eq. 6) employed for balancing the two models is designed solely to encourage better adherence to the provided spatial constraints. I will reconsider my rating if this aspect can be properly addressed.
- What is the purpose of initializing the coefficients as random values (Eq. 1)? This seems unnecessary and introduces random bias towards pixels.

**Limitations:**

The authors touched on the computational cost in Appendix B.5. However, a more detailed comparison with T2I and L2I models concerning computational overhead and memory overhead is crucial to determine the feasibility of the proposed method.

---

> ### Author Rebuttal · Authors · 2024-08-06
>
> *We sincerely thank you for your time and efforts in reviewing our paper, and your valuable feekback. We are glad to see that the proposed method is straightforward and flexible, the whole paper is well written, the idea is interesting and promising, and the experiments are extensive. Please see below for our responses to your comments.*
>
> **Q1: Since the proposed method runs multiple models concurrently and requires gradient-based updates, a discussion on computational efficiency is crucial for its practical application.**
>
> A1: Thank you for your suggestion. We compare our method with other T2I models and spatial-aware diffusion models regarding inference time, VRAM usage and complex performance in the table below:
>
> ||Training-free|Inference Time (/Img)|VRAM Usage|Complex$\uparrow$|
> |-|-|-|-|-|
> |SDXL|×|10.8s|14.7G|0.4091|
> |SD 3|×|10.4s|22.3G|0.3924|
> |Attn-Exct|√|14.3s|16.3G|0.3401|
> |ControlNet|×|11.2s|20.4G|-|
> |GLIGEN|×|16.8s|13.4G|0.3420|
> |LMD+ (SD 1.5)|√|26.6s|14.5G|0.3323|
> |RealCompo (SD 1.5 + GLIGEN)|√|23.8s|20.8G|**0.4657**|
>
> We observed that our method achieves better compositional generation results compared to other training-free approaches, with only marginal increases in inference time and VRAM usage. Our method can be flexibly applied to any T2I and spatial-aware diffusion models without the need for training.
>
> **Q2：Fig1 doesn't illustrate the trade-off between realism and compositionality well. I don't see an apparent drop of realism/aesthetics as $\beta$ increases.**
>
> A2: We are grateful for your feedback and apologize for any potential confusion caused. We have provided two more intuitive examples in **supplementary PDF**. As shown in **Fig3**, we conducted experiments using GLIGEN. We observed that as the layout control increased (with a higher $\beta$) or the number of layout control steps increased (with a higher $t_0$), the realism of the generated images declined. There is a noticeable degradation in both detail richness and aesthetic quality. For instance, the legs of the teddy bear appear unrealistic, as if it is facing backward with strange distortions, and the overall details of the rabbit become blurred and unappealing.
>
> Similarly, as shown in **Fig4**, we performed experiments using InstanceDiffusion, where we also define a parameter $\beta$ to control the strength of the layout control. It is evident that there is significant quality degradation in the dog's facial and body details. Additionally, the cat's eyes are different sizes, and the bird's legs are abnormally thin, indicating reduced realism in the generated images under the influence of layout control. **This suggests that achieving a balance between realism and compositionality in generated images is generally unattainable**.
>
> Thank you again for your feedback. We will update the manuscript to make Fig1 clearer and more intuitive.
>
> **Q3: Why does RealCompo improve image realism (Table 2, Fig6 left) and style preservation (Fig6 right) beyond the T2I model's capabilities? This is unclear, as the loss function (Eq6) aims only to enforce spatial constraints.**
>
> A3: First, we provide a detailed explanation of the concept of realism. Realism refers to the fidelity of details, aesthetic quality, and positional reasonableness of an image. Here, positional reasonableness is different from compositionality. Compositionality refers to the number of objects, spatial relationships, and attribute bindings in the generated images, **while positional reasonableness is an important metric for assessing the image realism**, specifically whether the placement of each object in the image is reasonable. **When the details and aesthetic quality of an image are similar, positional reasonableness becomes a key factor in user selection**. In Fig5 of the supplementary PDF, we provide examples from the user study, which demonstrates the advantages of RealCompo over the T2I model in realism. As shown in Fig5 (a), T2I model generates a teapot that is visibly suspended in the air, which doesn't conform to the physical laws of real-world scenes. In contrast, RealCompo generates objects within reasonable bounds through layout constraints, ensuring both the aesthetic quality and positional reasonableness. In Fig5 (b), the red chair generated by the T2I model is unnaturally placed on top of the table, and in Fig5 (c), two people generated by the T2I model are too close to each other. These examples illustrate that although T2I model outperforms in detail and aesthetics, its positional reasonableness needs improvement. Our method utilizes LLM to generate conditions that comply with physical laws, guiding the model to generate images with both high positional reasonableness and aesthetic quality. **Therefore, under similar detail and aesthetic quality, RealCompo's more reasonable and realistic composition gives it an advantage over the T2I model in terms of realism**.
>
> Second, the design of our loss function (Eq6) is crucial. As discussed in Sec 4.3 and illustrated in Fig8 of our paper, simply weighting the predicted noise from T2I and L2I models disrupts the object positions controlled by the L2I model, despite enhancing the detail and aesthetic quality of the generated images. This disruption occurs because T2I model lacks conditional controlled, leading to arbitrary positioning that interferes with layout control and results in uncontrollable compositionality. **Therefore, our loss function is essential to control the T2I model, ensuring it enhances image realism without compromising layout control**.
>
> **Q4: What is the purpose of initializing the coefficients as random values (Eq. 1)? This seems unnecessary and introduces random bias towards pixels.**
>
> A4: Thank you for your comment. Our fundamental goal is to ensure the initialization coefficients of the two models are the same, so it is indeed unnecessary to initialize them with random values. This is not the focus of our method, we will update and optimize this part in our paper.

---

> > ### Comment · Reviewer_juPc · 2024-08-11
> >
> > Thank the authors for the detailed response. I appreciate the additional information provided. However, I can't find the mentioned supplementary PDF that contains Figures 3, 4, and 5. I've checked the paper's appendix and the supplementary .zip file (which only has code). Could you please provide this PDF or clarify where to find it?

---

> > > ### Author Response · Authors · 2024-08-11
> > > **Response to Reviewer juPc**
> > >
> > > Dear Esteemed Reviewer juPc,
> > >
> > > Thank you for your kindly response. We would like to clarify that the supplement PDF refers to **the pdf in global response**. We apologize for any confusion this may have caused.
> > >
> > > Should there be any further points that require clarification or improvement, please know that we are fully committed to addressing them promptly.
> > >
> > > Thank you once again for your invaluable contribution to our research.
> > >
> > > Warm Regards, The Authors

---

> > > > ### Comment · Reviewer_juPc · 2024-08-12
> > > >
> > > > Thank the authors for the clarification.
> > > >
> > > > I would like to believe the explanation about realism improvement. However, the improved aesthetic quality, as indicated by Table 2 and the second paragraph of Sec 4.2, is still hard to comprehend. Meanwhile, the authors also noticed that "T2I model outperforms in detail and aesthetics" in Fig 5 in the newly provided PDF. I would like the authors to further clarify this.

---

> > > > > ### Author Response · Authors · 2024-08-12
> > > > > **Response to Reviewer juPc**
> > > > >
> > > > > Thank you for the response! In the following we provide additional explanations regarding to your questions. Please feel free to let us know if these address your concerns.
> > > > >
> > > > > **Q1: The improved aesthetic quality, as indicated by Table 2 and the second paragraph of Sec 4.2, is still hard to comprehend.**
> > > > >
> > > > > A1: We would like to emphasize that the aesthetic predictor was trained using the Aesthetic Visual Analysis (AVA) and Simulacra Aesthetic Captions (SAC) datasets, **both of which are annotated by experts or photography enthusiasts**. In these datasets, each image is given a score ranging from 1 to 10 based on its perceived appeal. **It is important to note that these annotators tend to be more critical of images that appear unnatural or fake (i.e., images with unrealistic or awkward compositions)**. The aesthetic predictor is designed to filter out high-quality images that excel in both detail representation and composition. The aesthetic scores for the images in Figure 5 in the newly provided PDF are provided as follows:
> > > > >
> > > > > |                    | Aesthetic Score$\uparrow$ |
> > > > > | ------------------ | ----------------------------- |
> > > > > | Fig5 (a)-T2I Model | 5.304                         |
> > > > > | Fig5 (a)-RealCompo | **5.653**                     |
> > > > > | Fig5 (b)-T2I Model | 5.269                         |
> > > > > | Fig5 (b)-RealCompo | **5.775**                     |
> > > > > | Fig5 (c)-T2I Model | 5.471                         |
> > > > > | Fig5 (b)-RealCompo | **5.884**                     |
> > > > >
> > > > > From the table, it can be observed that when the images have similar levels of detail and visual refinement, **the aesthetic predictor assigns a higher aesthetic score to images with more reasonable positioning**.
> > > > >
> > > > > **Q2: The authors also noticed that "T2I model outperforms in detail and aesthetics" in Fig 5 in the newly provided PDF.**
> > > > >
> > > > > A2: Thank you for your comment, and we apologize for any confusion caused. Here, we intended to convey that T2I models excel in terms of detail and **visual refinement**. We realize that our use of the term "aesthetic" may have led to confusion, as **it was not meant to refer to the aesthetic score in this context**. We apologize again for any misunderstanding this may have caused.
> > > > >
> > > > > We will review and refine the manuscript to to clarify the two issues you have raised.

---

> > > > > > ### Comment · Reviewer_juPc · 2024-08-12
> > > > > >
> > > > > > I would like to thank the authors for their detailed explanations. Having considered the responses to my questions and those of other reviewers, I am changing my rating from borderline reject to weak accept.

---

> > > > > > > ### Author Response · Authors · 2024-08-12
> > > > > > > **Thank you for your support**
> > > > > > >
> > > > > > > Thank you very much for raising the score! We sincerely appreciate your valuable comments and the time and effort you put into reviewing our paper. We will make sure to incorporate these suggestions into the final manuscript.
> > > > > > >
> > > > > > > Warm Regards,
> > > > > > >
> > > > > > > The Authors

---

### Official Review · Reviewer_XVVX · 2024-07-13

**Soundness:** 3
**Presentation:** 3
**Contribution:** 3
**Rating:** 7
**Confidence:** 5

**Summary:**

The paper introduces a training-free and flexible text-to-image generation framework called RealCompo, which enhances compositional text-to-image generation by balancing the realism and compositionality of generated images. It features a novel balancer that dynamically combines the predicted noise from T2I models and spatial-aware image diffusion models (such as layout, keypoint, segmentation map). This framework provides a fresh perspective for compositional image generation.

**Strengths:**

1. The proposed method is training-free but achieves notable results.
2. The proposed method mainly applies a balance between two pretrained diffusion models, which is easy to implement and quite flexible.
3. The paper’s explanation of realism and control sounds reasonable.

**Weaknesses:**

1. The biggest problem is novelty/contribution. The paper uses a balancer to balance two diffusion models, which is very similar to the concept of "checkpoint merge" / "model ensemble". It is necessary to clarify the differences between these two approaches.

**Questions:**

I will consider increasing the score or maintaining it if the authors can address the following issues.
1. The paper uses a balancer to balance two diffusion models, which is very similar to the concept of "checkpoint merge" or "model ensemble". It is necessary to clarify the differences between these two approaches.
2. Please explain the necessity of the proposed method. If condition-guided models like ControlNet use a stronger backbone, higher quality, larger quantity, and more diverse data and annotations during training, is it still necessary to balance with non-condition-guided text-to-image models?

**Limitations:**

Yes.

---

> ### Author Rebuttal · Authors · 2024-08-06
>
> *We sincerely thank you for your time and efforts in reviewing our paper, and your valuable feekback. We are glad to see that the proposed method is flexible, the experimental results are promising, and the explanation of the important concepts is reasonable. Please see below for our responses to your comments.*
>
> **Q1: The paper uses a balancer to balance two diffusion models, which is very similar to the concept of "checkpoint merge" or "model ensemble". It is necessary to clarify the differences between these two approaches.**
>
> A1: Thank you for the comment. We here clarify the differences between our method and checkpoints merge and model ensemble.
>
>  Checkpoint merge aims to combine parameters from multiple models; however, this method presents a problem in compositional generation. Checkpoint merge is **typically used to blend the realism or stylistic features of two models**, requires the model to possess a high level of spatial awareness and precise localization capabilities. Simply merging the parameters of two models, such as SD and GLIGEN, may **compromise the strength of layout control, leading to a decline in compositionality**. Our method merges the predicted noise from both models, as **each contains unique strengths in either rich semantic information or spatial features**. This fusion method more effectively leverages the strengths of both models.
>
> Model ensemble, on the other hand, involves a straightforward weighted combination of the output noises from two models. We have discussed the limitations of this approach in our paper. As seen **in Section 4.3 and Figure 8**,  although simply weighting the predicted noise from T2I and L2I models can result in a generated image with higher realism, **it disturbs the object positions of the L2I model**. The lack of conditioning in the T2I model leads to arbitrary object placements, causing conflicts with the layout's object positions and resulting in uncontrollable composition. Thus, model ensemble has limitations in compositional generation, and **dynamic balancer is a crucial bridge** for balancing the predicted noise from the models.
>
> Therefore, in compositional generation tasks, both checkpoint merge and model ensemble methods have their own shortcomings when it comes to achieving a trade-off between realism and compositionality.
>
> **Q2: Please explain the necessity of the proposed method. If condition-guided models like ControlNet use a stronger backbone, higher quality, larger quantity, and more diverse data and annotations during training, is it still necessary to balance with non-condition-guided text-to-image models?**
>
> A2: Even though condition-guided models use more powerful backbones with larger sizes and higher performance, **the challenge of achieving a balance between realism and compositionality in generated images still exists**. Therefore, our approach is necessary and  widely needed in the field of controllable generation. In the **supplementary PDF**, we provide additional experimental results to validate the necessity of our method. As shown in Figure 1, we conducted experiments using layout-based models (GLIGEN) of different sizes, varying the parameter $\beta$ to control the strength of condition guidance. It is evident that the aesthetic quality of the generated results significantly decreases as the condition guidance strength increases, **even with a larger and stronger backbone**. This indicates that improving compositionality comes at the expense of realism. Similarly, in Figure 2, we conducted experiments using keypoint-based models (ControlNet) of different sizes, varying the parameter *control_scale* to control the strength of condition guidance. Again, it is apparent that larger and stronger backbones, trained with more and better data, exhibit superior realism under the same conditions. **However, as the strength of the condition control increases, a notable decline in realism is observed**. Therefore, **the trade-off between realism and compositionality is a common issue**.

---

> > ### Author Response · Authors · 2024-08-11
> > **A minor clarify**
> >
> > Dear Esteemed Reviewer XVVX,
> >
> > We would like to clarify that the supplement PDF refers to **the pdf in global response**. We apologize for any confusion this may have caused.
> >
> > Should there be any further points that require clarification or improvement, please know that we are fully committed to addressing them promptly.
> >
> > Thank you once again for your invaluable contribution to our research.
> >
> > Warm Regards,
> >
> > The Authors

---

> ### Comment · Reviewer_XVVX · 2024-08-12
> **please plot the coefficent curve**
>
> 1. It's recommended to plot the current dynamic coefficient curve.
> 2. Is there exisits a general coefficient curve that is suitable for most of the prompt? If it is possible, there's no need for an optimization-based method that requires gradient backward, which will be mush easier for real application.
> 3. It's recommended to show the results of the fixed coefficient (the best selected) instead of dynamic coefficient.

---

> > ### Author Response · Authors · 2024-08-12
> > **Response to Reviewer XVVX**
> >
> > Thank you for the response! In the following we provide additional explanations regarding to your questions. Please feel free to let us know if these address your concerns.
> >
> > **Q1: It's recommended to plot the current dynamic coefficient curve.**
> >
> > A1: Thank you for your suggestion. Unfortunately, **due to NeurIPS's requirement that no anonymous links be included in the rebuttal**, we regret that we cannot provide you with the dynamic coefficient curve at this time. However, we will certainly include this experiment and its analysis in the revised manuscript. Just a gentle reminder, **in Figure 10 of our paper**, we present the curve showing the gradient of the loss function (Eq 6) with respect to the coefficient during the denoising process. The figure illustrates that the coefficient exhibits significant convergence during denoising, and there are distinct update patterns for the coefficient when using different backbones in RealCompo.
> >
> > **Q2: Is there exisits a general coefficient curve that is suitable for most of the prompt? If it is possible, there's no need for an optimization-based method that requires gradient backward, which will be mush easier for real application.**
> >
> > A2: It is a valuable view to explore a general coefficient curve. However, we observed that the coefficient varies depending on the prompt based on extensive experiments. This is because **the coefficient is primarily optimized based on the layout, and different prompts correspond to different layouts, leading to varying coefficients**. We will certainly include visualizations and analysis of this part in the revised manuscript. Additionally, as seen in Figure 10, **RealCompo with different backbones also requires distinct coefficient update strategies**.
> >
> > But I agree that having a general coefficient curve would make the application of our method more convenient and meaningful. Thank you for your thoughtful suggestions. This is a preliminary attempt, and we will explore the potential of a general coefficient curve in the future.
> >
> > **Q3: It's recommended to show the results of the fixed coefficient (the best selected) instead of dynamic coefficient.**
> >
> > A3: Thank you for your suggestion. **In the third column of Figure 8** (w/o Dynamic Balancer) in the paper, we present the results of experiments **using a simple fixed coefficient, where both models have the same coefficient**. The figure illustrates that without dynamically updating the coefficient, the T2I model, which lacks layout constraints, negatively impacts the positioning capability of the L2I model. This results in generated images where the object positions do not align with the layout, despite the T2I model retaining a higher realism advantage. We will include additional manually designed fixed coefficients in the manuscript to validate the effectiveness of our method.

---

> > > ### Comment · Reviewer_XVVX · 2024-08-13
> > > **The rebuttal addressed my concerns.**
> > >
> > > 1. The rebuttal has satisfactorily addressed my concerns regarding real-world applications, particularly the issue that the proposed method requires gradient backpropagation, which demands more memory and increases inference time. The training-free approach introduced in this paper achieves remarkable improvements in both controllability and realism. It also expands the family of model ensembling/checkpoint merging techniques, which are extensively used in the diffusion community. Considering that classifier-free guidance also requires double the forward passes but results in exceptional performance, I am inclined to view this paper positively.
> > > 2. Given the potential of this paper to apply a fixed coefficient that eliminates the need for gradient backpropagation, I am raising my score from weak accept to accept. Nevertheless, I strongly recommend that the authors provide further analysis on the use of a fixed coefficient. For example, in real-world applications, although we can apply an optimal or dynamic CFG for different models, prompts, or timesteps, a fixed constant CFG strategy is often employed and still results in significant improvements. Additionally, a fixed dynamic CFG for different timesteps that does not require gradient backpropagation, such as cosine CFG or the CFG used by MDT, could be considered.
> > > 3. Leveraging large language models (LLMs) to orchestrate various tools and utilize multiple models to complete complex tasks—such as generating layouts, keypoints, or segmentation as intermediate steps—represents a promising future direction. This approach could provide more effective solutions for challenging problems like multi-object generation and attribute binding.

---

> > > > ### Author Response · Authors · 2024-08-13
> > > > **Thank you for your support**
> > > >
> > > > Thank you very much for raising the score! We sincerely appreciate your valuable comments and the time and effort you put into reviewing our paper.
> > > >
> > > > Based on your thoughtful suggestion of a fixed-coefficient strategy, we make sure to plan additional experiments to analyze its feasibility and identify the optimal fixed coefficient to enhance the efficiency and quality of our method.
> > > >
> > > > Thank you once again for your invaluable contribution to our research.
> > > >
> > > > Warm Regards,
> > > >
> > > > The Authors

---

### Author Rebuttal · Authors · 2024-08-06

We sincerely thank all the reviewers for the thorough reviews and valuable feedback. We are glad to hear that the idea is interesting, promising and flexible (Reviewer juPc), the paper is well written and easy to follow (Reviewer XVVX, juPc),  the experiments are extensive and performance improvements are promising (all reviewers).

To address the reviewers' concerns and misunderstandings, we would like to emphasize the contributions and novelty of our method by highlighting two key points that distinguish our work from previous research and underscore the significance of our findings.

***First, we discovered for the first time the imbalance between realism and compositionality of generated images in compositional generation. Our study provides experimental analysis and conclusions on this issue.***

Previous controllable generation methods often overlooked the decline in image realism while focusing on enhancing compositionality. We introduce a critical new challenge to the field of controllable generation: achieving a balance between the realism and compositionality of generated images. In Figure 1 of the paper and Figures 1, 2, 3, and 4 of the supplementary PDF, we conduct detailed analyses using multiple models and experiments to identify the root causes of this imbalance.

***Second, we provide a new perspective on optimization-based generation methods, achieving satisfactory results and generalization by using coefficient updates for the first time.***

Previous optimization-based generation methods have focused on optimizing the latent $z_t$ based on the loss function. However, this approach is unsuitable for our task, which requires dynamically combining the strengths of two models. Therefore, we introduce a novel perspective to address this challenge by dynamically optimizing the coefficients of the models. This approach effectively leverages the advantages of each model, achieving a balance between realism and compositionality. Additionally, due to the flexibility and generalizability of our method, we can seamlessly integrate any stylized T2I models and spatial-aware diffusion models to achieve high-quality stylized compositional generation.

We here summarize and highlight our responses to the reviewers:

- We thoroughly emphasize our contributions and novelty (Reviewer XVVX, juPc and ZHfz), and clarify the differences between our method and "checkpoints merge" or "model ensemble (Reviewer XVVX)."
- We conduct extensive experiments on different sizes of different backbones to verify that the issue of imbalance between realism and compositionality in controllable generation is widespread (Reviewer XVVX, ZHfz). Additionally, we provided more clear and intuitive examples to explain the motivation of the paper as illustrated in Fig1 in detail (Reviewer juPc, ZHfz).
- We provided a more detailed explanation of realism and offered a nexperimental analysis of why our method can prevent the generation of unrealistic images (Reviewer juPc and ZHfz).
- We compare our method with other diffusion models regarding inference time, VRAM usage and complex performance, achieving a better trade-off than previous diffusion models (Reviewer juPc).


We reply to each reviewer's concerns in detail below their reviews. Please kindly check out them. Thank you and please feel free to ask any further question.

---

### Decision · Program_Chairs · 2024-09-25

**Decision:**

Accept (poster)

**Comment:**

All the reviewers recommend acceptance of the work. Reviewers appreciated the simple yet effective technique with good results. Reviewers raised several concerns related to novelty and clarification questions, several of which are addressed in the rebuttal. Reviewers did raise valid concerns and authors are encouraged to do the best of their abilities to include the rebuttal discussions and address all the concerns in the final version.